# Influenza A virus ribonucleoproteins form liquid organelles at endoplasmic reticulum exit sites

Marta Alenquer [1], Sílvia Vale-Costa[1], Temitope Akhigbe Etibor[1], Filipe Ferreira[1], Ana Laura Sousa[1,2] & Maria João Amorim [1]

Influenza A virus has an eight-partite RNA genome that during viral assembly forms a complex containing one copy of each RNA. Genome assembly is a selective process driven by RNA-RNA interactions and is hypothesized to lead to discrete punctate structures scattered through the cytosol. Here, we show that contrary to the accepted view, formation of these structures precedes RNA-RNA interactions among distinct viral ribonucleoproteins (vRNPs), as they assemble in cells expressing only one vRNP type. We demonstrate that these viral inclusions display characteristics of liquid organelles, segregating from the cytosol without a delimitating membrane, dynamically exchanging material and adapting fast to environmental changes. We provide evidence that viral inclusions develop close to endoplasmic reticulum (ER) exit sites, depend on continuous ER-Golgi vesicular cycling and do not promote escape to interferon response. We propose that viral inclusions segregate vRNPs from the cytosol and facilitate selected RNA-RNA interactions in a liquid environment.

[1] Cell Biology of Viral Infection Lab, Instituto Gulbenkian de Ciência, 2780-156 Oeiras, Portugal. [2] Electron Microscopy Facility, Instituto Gulbenkian de Ciência, 2780-156 Oeiras, Portugal. Equal contribution: Marta Alenquer, Sílvia Vale-Costa Correspondence and requests for materials should be addressed to M.J.A. (email: mjamorim@igc.gulbenkian.pt)

nfluenza A infections are serious threats to human health, causing annual epidemics, and occasional pandemics[1]. The virus contains an eight-partite RNA genome, with each segment encapsidated as an individual viral ribonucleoprotein (vRNP) complex. vRNPs are composed of single-stranded negative-sense RNA, with base paired terminal sequences originating a double-stranded RNA portion to which binds the trimeric RNA-dependent RNA polymerase (RdRp), composed of PB1, PB2, and PA. The remaining sequence attaches several copies of unevenly-bound nucleoprotein (NP)[2]. The advantages of having a segmented genome are evident for viral evolution[3] and for better gene expression control[4], but increase the complexity of the assembly of fully infectious virions[5,6].

Viral assembly occurs at the plasma membrane. For an influenza particle to be fully infectious, the eight vRNPs must be packaged in a virion. Virions do not usually package more than eight segments[7] and each segment generally occurs once per virion. In agreement, full-length segments compete with corresponding defective interference particles (segments that have internal deletions)[8–10]. Together, the data indicate that vRNP segments of the same type do not interact. At the budding sites, complexes of eight interlinked vRNPs have been imaged, meaning that, at some point during infection, the eight segments establish specific *cis*-acting and intersegment interactions to form a supra-molecular complex[5,6]. However, it is under debate whether vRNPs reach the plasma membrane already as complete genome bundles.

Upon exiting the nucleus, where they replicate, vRNPs accumulate around the microtubule organizing centre[11]. Subsequently, vRNPs distribute throughout the cytoplasm, concentrating in discrete puncta that enlarge as infection progresses[11–15]. Each puncta accommodates different vRNP segments with the diversity in vRNPs increasing with proximity to the plasma membrane[15]. These observations led to the proposal that genome assembly precedes vRNP packaging into budding virions by a process linked with the formation of the referred vRNP hotspots[12,14–16]. Studies on the biogenesis of vRNP hotspots showed that their formation required the cellular GTPase Rab11[11,13,17,18]. In uninfected cells, Rab11 is the master regulator of the endocytic recycling compartment (ERC), a system used for delivering endocytosed material and specific cargo from the *trans*-Golgi network (TGN) to the cell surface. Rab11-GTP regulates ERC transport by recruiting molecular motors, tethers, and SNARES to, respectively, drive, dock, and fuse vesicles to the plasma membrane[19]. Despite initial reports showing that the role of Rab11 was to deliver vRNPs to the cell surface[11,13,18,20], accumulating evidence strongly indicates that Rab11 subcellular localization is redirected and its function is impaired during IAV infection[14,17,21]. In fact, it was demonstrated that vRNPs out-compete Rab11 effectors for Rab11 binding, rendering the recycling process sub-optimal[14,21]. Further corroborating the scenario that Rab11 pathway is impaired by infection, a recent publication showed that Rab11 was re-routed to the ER during IAV infection[17]. In addition, Rab11 is redistributed during infection, changing from discrete to enlarged puncta that match sites of clustered vesicles positive for Rab11 and vRNPs, constituting vRNP hotspots[14].

The formation of vRNP hotspots was postulated to be dependent on the establishment of sequential RNA–RNA interactions occurring as Rab11 vesicles transporting vRNPs collided[12,15]. However, impairment of endocytic recycling argues against this hypothesis and challenges the IAV assembly model proposed. Nevertheless, the existence of vRNP/Rab11 hotspots indicates segregation from the cytosol in foci. Non-classical organelles, in the sense that they are not delimitated by membranes, are abundant in the viral world and are known as viroplasms, viral factories,

aggresomes, or virosomes, to indicate sites of viral replication[22,23]. Viruses can also form viral inclusions, which are sites of accumulation of viral proteins, nucleic acids and selected host proteins and can include (or not) viral factories[22]. Given this definition, IAV vRNP hotspots could be re-classified as viral inclusions. The most notable cases of segregated material from the cytosol (not delimited by membranes) are found in cells infected by viruses of DNA (*Poxviridae, Iridoviridae, Asfaviridae*), of dsRNA (*Reoviridae*) and of negative-sense RNA (*Paramyxoviridae, Rhabdoviridae, Filoviridae*) genome[22–25]. Formation of factories is associated with remodeling of host membranes and/or cytoskeleton to orchestrate sophisticated platforms for viral replication and/or for escaping host immune recognition[23]. However, several questions remain unclear relative to the internal organization and biophysical properties of these cellular condensates. Resolving these questions for IAV will help to understand how viruses organize matter into membraneless organelles and identify their functions.

Here, we disclose that vRNP/Rab11 hotspots constitute viral inclusions that are not delimited by membranes and display characteristics of liquid organelles. Liquid properties include dynamic change of components, round appearance, easy deformation, fusion/fission events, and fast adaptation to physiological changes. We also show that viral inclusion formation is spatially regulated, assembling in the vicinity of the Endoplasmic Reticulum Exit Sites (ERES), and depends on continuous ER-Golgi vesicular cycling. We demonstrate that viral inclusions are formed when a single vRNP type is expressed in a cell. Therefore, contrary to the most accepted view, these sites are not formed by established intersegments interactions. Importantly, we show that viral inclusions accommodate segments from different parental strains in a co-infection system and do not promote escape from the interferon (IFN) antiviral response. We propose that viral inclusion formation precedes and facilitates stochastic vRNP–vRNP interactions in a liquid environment of crowded vRNPs.

## Results

**IAV inclusions form when a single vRNP type is expressed**. The most disseminated model for IAV genome assembly hypothesizes that genome transport is coupled to genome assembly[15,26,27]. The model predicts that transiently colliding vRNP-carrying vesicles promote the establishment of intersegment interactions. Experimental support for this model includes microscopy-based approaches showing co-localization of different vRNP types[12,15] and Rab11[11] in viral inclusions, which enlarge with time of infection[14]. A question in the field is, therefore, whether intersegment interactions drive formation of viral inclusions. To answer this question, we assessed the formation of vRNP hotspots and the subcellular distribution of Rab11 in a mini-replicon system expressing a single segment type, segment 7 of influenza A/Puerto Rico/34/8 (PR8). As stated in the introduction, it was demonstrated that vRNPs from the same type compete for packaging in virions, and, therefore, do not interact[10,28,29]. As positive control, we used cells expressing two PR8 vRNPs, segments 7 and 8, previously shown to form viral inclusions[11]. Cells were transfected with plasmids expressing the RdRp, a 4:1 mixture of NP:GFP-NP (3P-NP), NS2 (to ensure nuclear export of vRNPs), segment 7 (that encodes for M1 and M2) and, when indicated, segment 8 (that expresses NS1 and NS2). Segment transcription originates a complete negative-sense RNA to which NP and the RdRp bind, amplifying the system and mimicking viral transcription and replication. As negative control, the same system without the polymerase PB2 was evaluated (2P-NP). This mini-replicon has been validated before, by showing that vRNPs incorporated GFP-NP, and that complete

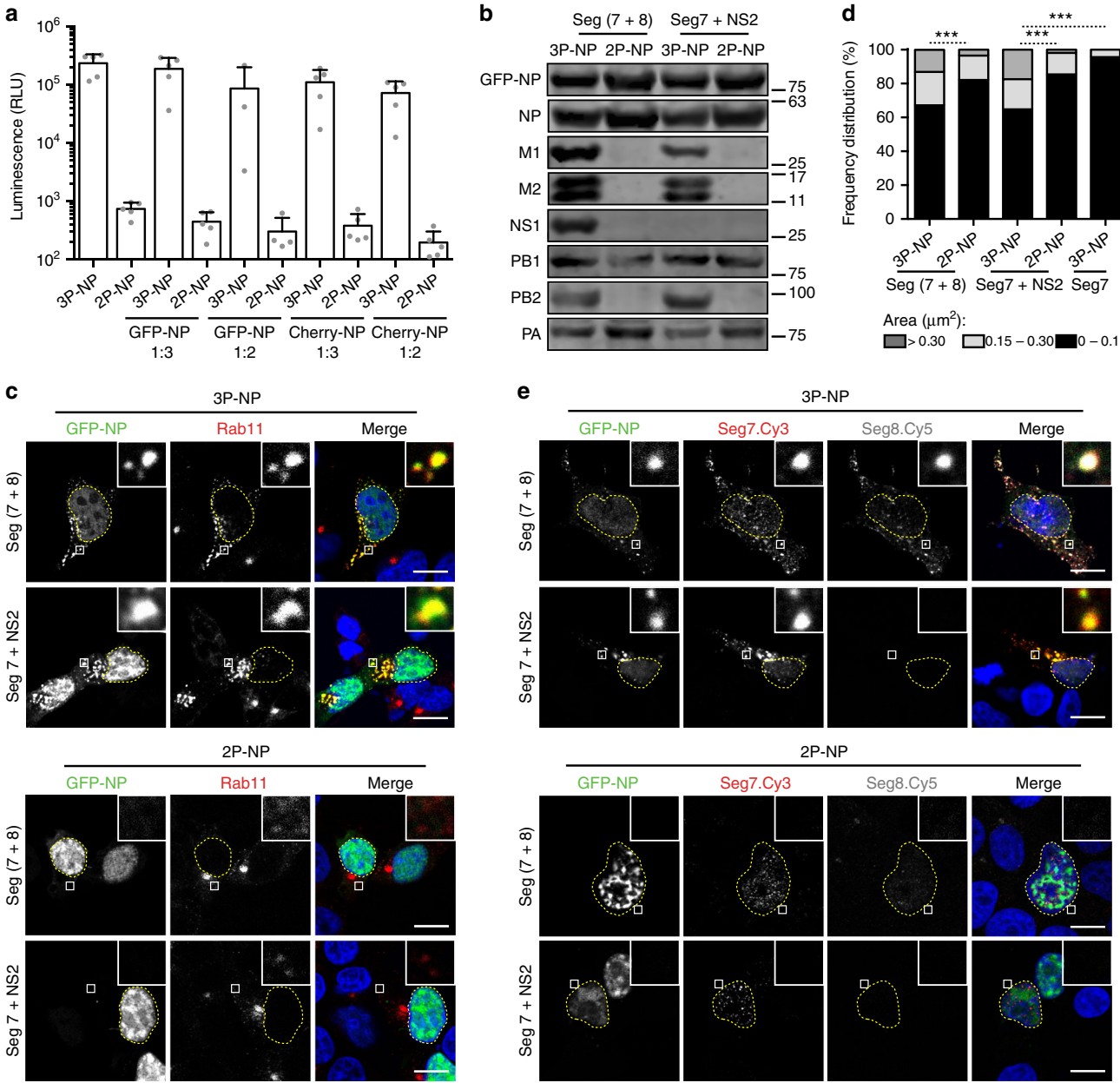

**Fig. 1** Viral inclusions form in the absence of intersegment RNA–RNA interactions. 293 T cells were transfected for 16 h with the minimal protein components of an influenza vRNP: the three polymerase proteins (3P) (or, as a nonfunctional control, two polymerase proteins lacking PB2 - 2P) and NP, as well as with plasmids expressing GFP-NP and **a** a luciferase reporter plasmid cloned in negative sense and flanked by influenza promoter. mCherry-NP was also used instead of GFP-NP, when indicated. Luminescence was determined for luciferase expression and the values plotted as mean ± standard error of the mean (SEM). Results are a pool of 2 independent experiments; or **b**–**e** 293 T cells were further transfected with plasmids expressing vRNA from segments 7 and 8, or segment 7 alone, and a plasmid encoding NS2 when segment 7 was expressed alone. **b** Cells were lysed and the indicated proteins were detected by western blotting. **c** Cells were fixed and stained for Rab11 (red). White boxes show areas of co-localization between NP and Rab11. Nuclei are delineated by yellow dashed lines. Bar = 10 μm. **d** The frequency distribution of Rab11 inclusions within the three area categories (in μm$^2$) was plotted for each condition. Statistical analysis of data was performed using a non-parametric Kruskal–Wallis test, followed by Dunn's multiple comparisons test (***$p < 0.001$). Between 30 and 70 cells were analyzed per condition and 3 independent experiments were performed. **e** Duplicate samples were processed to detect segment 7 (red) and segment 8 (gray) RNA by FISH. White boxes show areas of co-localization between NP and viral segments. Nuclei are delineated by yellow dashed lines. Bar = 10 μm

vRNPs were exported from the nucleus, colocalizing with RNA from all segments[11].

In the present study, we further confirmed that the system was fully functional by several methods. First, we certified that GFP-NP or mCherry-NP transfection supported the polymerase function in the conditions assayed, using a mini-genome reporter plasmid that produces a negative-sense luciferase gene under the control of viral promoter sequences in cells expressing 3P-NP/GFP-NP or mCherry-NP, in relation to the 2P-NP/GFP-NP or mCherry-NP negative controls (Fig. 1a). Subsequently, the expression of all components of the mini-replicon, in each condition, was evaluated by western blotting, except that of NS2,

for which no good commercial antibody is available (Fig. 1b). As expected, the expression of the proteins of a specific segment was detected only in 3P-NP samples (Fig. 1b). We then investigated Rab11 subcellular distribution by immunofluorescence, and observed that it did not change in any of the 2P-NP conditions, consistently with previous reports (Fig. 1c, lower panels)[11]. However, in the 3P-NP condition, Rab11 redistributed, forming the characteristic enlarged puncta regardless of expressing one or two vRNPs, indicating that one vRNP type is sufficient to form viral inclusions (Fig. 1c, upper panels). The areas of Rab11 puncta, when one or two vRNPs were expressed, were significantly different between the 3P-NP and 2P-NP conditions, when quantified, and ranked based on their size: small inclusions up to $0.15\,\mu m^2$, intermediate inclusions between 0.15 and $0.30\,\mu m^2$, and large inclusions over $0.30\,\mu m^2$ (Fig. 1d), as before[14]. Consistent with our work, in the absence of NS2, vRNPs were retained in the nucleus and Rab11 distribution was similar to that of 2P-NP condition (Fig. 1d, 3P-NP seg7 without NS2). Finally, we assessed the distribution of segment 7 and 8 RNA by fluorescent in situ hybridization (FISH). In the case of 2P-NP conditions, probes against the vRNA of segments 7 or 8 detected discrete dots in the nucleus (Fig. 1e, lower panels), consistent with DNA polymerase I transcription but absence of amplification, as described before[11]. In the case of 3P-NP, and independently of the number of segments expressed, vRNAs of two types were detected in enlarged puncta, and co-localized between them and with NP, showing that vRNP hotspots are formed without requiring RNA interactions among distinct segments (Fig. 1e, upper panels).

Collectively, the results obtained demonstrate that viral inclusions assemble in the presence of a single vRNP type. The data also indicate that formation of Rab11 enlarged puncta is dependent on vRNPs reaching the cytosol, but precedes and does not require RNA–RNA intersegment interactions.

**Viral inclusions display properties of liquid organelles**. Our previous result suggests that formation of viral inclusions containing vRNPs and the host protein Rab11 precedes the establishment of intersegment interactions. Recently, biomolecular condensates of nucleic acids and proteins, such as nuclear speckles, nucleolus, centrioles, and stress granules, were shown to form by liquid–liquid phase separation and to have liquid properties[30]. Liquid-like properties include deformability promoted by fusion and fission events, quick adaptation to stimuli, dynamic exchange of material, ability to internally reorganize and rounded shape[30,31]. Similar properties were shown for Negri bodies and viroplasms formed during rabies and vesicular stomatitis viral infections[24,32], and it was postulated that other viral factories or viral inclusions would assemble by liquid–liquid phase separation.

To test this idea for influenza A viral inclusions, we used two different systems with tagged vRNPs for monitoring their dynamic nature and properties in live cells: (1) a productive PR8 virus encoding PA-GFP in its genome that replicates at the same levels as the wild type (WT) virus (Fig. 2a), described by Bhagwat et al.[33], and (2) simultaneous transfection with GFP-NP and infection with PR8 (GFP-NP/PR8 system) that was shown to produce full-length vRNPs incorporating GFP-NP[34]. We first confirmed that viral inclusions were composed of similar components to those of the WT viruses. As reported by us and others[11,12,15], cells infected with PR8 at 16 h post-infection (hpi) produced cytosolic viral inclusions containing vRNPs, as shown by the co-localization between NP and segment 1 and 3 RNAs (Fig. 2c, left upper panel). The same was observed for cells transfected with control GFP (Fig. 2c, right upper panel) or GFP-

NP/PR8 (Fig. 2c, right lower panel). Co-localization between segment 3 RNA and PA protein of the virus encoding PA-GFP was also detected (Fig. 2c, left lower panel). As a final step in the characterization of viral inclusions, we combined the two systems (simultaneous mCherry-NP transfection and PA-GFP virus infection) and observed the development of similar viral inclusions that where both PA-GFP and mCherry-NP positive (Fig. 2b, Supplementary Movie 1). Together, data indicate that PA-GFP encoding virus and GFP-NP/PR8 systems behave as WT PR8 virus in terms of producing cytosolic viral inclusions composed of PA and RNA or NP and RNA, respectively.

Subsequently, we analyzed dynamic events of individual viral inclusions in the two systems. Careful analyses revealed many fusion events amongst individual inclusions either separated by short or long distances, and also fission events (Fig. 2d, e, Supplementary Fig. 1, Supplementary Movies 2 and 3), which indicates constant exchange of material and deformability.

To check if the abovementioned events resulted from association with molecular motors and the cytoskeleton, we treated cells with nocodazole and latrunculin A, drugs that depolymerize microtubules and actin, respectively. Although nocodazole greatly reduced fusion and fission events, latrunculin A treatment did not have an effect on either event for both viral infections (Fig. 2f, g, Supplementary Movies 4-9). Interestingly, upon nocodazole treatment, viral inclusions localized close to the plasma membrane and were reduced in number (Supplementary Movies 5 and 8). Of note, in untreated samples upon fusion/fission events, viral inclusions reacquired a rounded shape, indicating occurrence of internal rearrangements. Some acquisition of material originated from several compartments, being difficult to track their origin (Supplementary Fig. 1c, d, complex, Supplementary Movies 2 to 4, 7, 23 and 25) and in most viral inclusions a constant flux of small material in and out was observed (Supplementary Fig. 1a, b, arrows, Supplementary Movies 4 and 24). Furthermore, these inclusions appear to be constantly exchanging material amongst them, an essential trait if these were sites devoted to viral genome assembly.

**Viral inclusions react promptly to physiological changes**. To investigate how viral inclusions react to physiological changes, we subjected them to hypotonic shock. Infected cells with vRNPs incorporating PA-GFP or GFP-NP were live-imaged by confocal microscopy. After ~1 min, cells were subjected to a rapid hypotonic shock (ionic strength changing from 150 to 30 mM by diluting media with water). Viral inclusions, otherwise stable over time, immediately started to dissolve, and 2 min later were no longer visible in both systems (Fig. 3a, b Supplementary Movies 10, 11, 13 and 14). Quantifications were done in fixed cells stained for NP, and the percentage of cells with viral inclusions was plotted (Fig. 3c). Hypotonic stress resulted in almost complete dissolution (from $88.1 \pm 5.0\%$ to $6.4 \pm 8.1\%$ for PR8; $82.0 \pm 8.0\%$ to $7.1 \pm 12.6\%$ for PA-GFP; $80.6 \pm 5.8\%$ to $4.5 \pm 9.6\%$ for GFP-NP/PR8, average percentage $\pm$ SD) of viral inclusions after 10 min of treatment. This effect was reversible as upon substitution of hypotonic solution by normal media, followed by 1 h incubation, viral inclusions reformed ($91.0 \pm 2.7\%$ for PR8; $76.1 \pm 8.2\%$ for PA-GFP; $75.8 \pm 6.5\%$ for % for GFP-NP/PR8).

The hypotonic shock effect was confirmed in cells infected with PR8, fixed and stained with NP and Rab11 or vRNA. Mock-infected cells only display Rab11, with no effect of hypotonic shock detected (Supplementary Fig 2a, b). In infected cells, vRNA, NP and Rab11 were all sensitive to 10 min of hypotonic shock and re-aggregated after re-incubation with media for 1 h (Supplementary Fig 2a, b). In sum, the data indicate that cytosolic viral inclusions react quickly to stress.

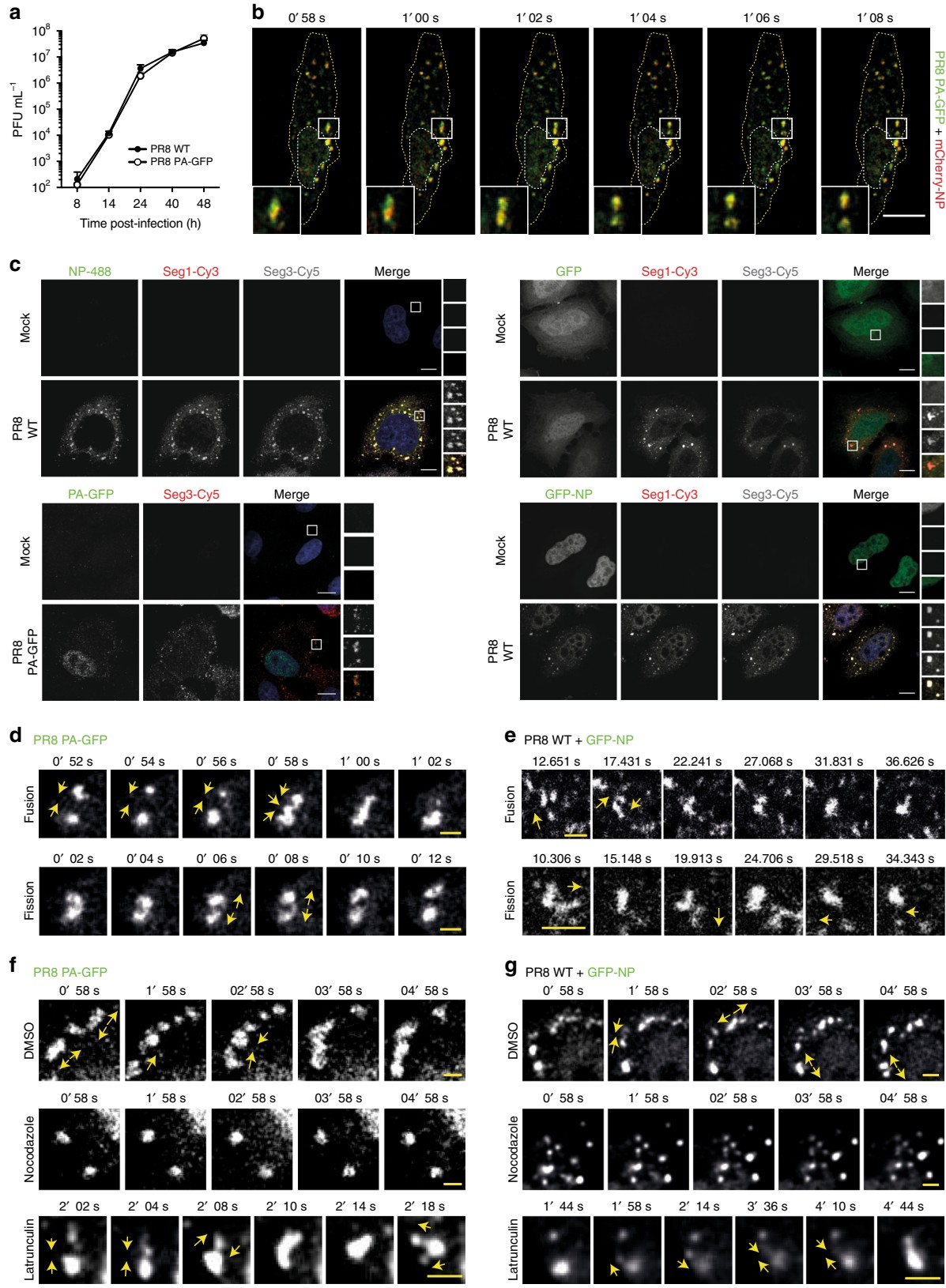

Biomolecular condensates exhibit diverse physical properties, ranging from liquid-like to solid-like behaviors. Liquid- and solid- like condensates are morphologically similar, but can be differentiated by careful usage of the aliphatic alcohol 1,6-hexanediol[35]. We observed that viral inclusions are sensitive to a 5–10 min treatment with 5% hexanediol, for both PA-GFP virus and GFP-NP/PR8 system, as tested by live-cell imaging (Fig. 3a, b, Supplementary Movies 12 and 15). Interestingly, inclusions did not dissolve at the same time in the same cell, and a small percentage even resisted to the treatment, which indicates that

**Fig. 2** PA-GFP virus and GFP-NP/PR8 system form dynamic cytosolic viral inclusions. **a** A549 cells were infected at an MOI of 0.001 with PA-GFP encoding PR8 virus or a WT virus to follow their growth over 48 h on a multicycle assay. **b** A549 cells were transfected with a plasmid encoding mCherry-NP and co-infected with PA-GFP virus at an MOI of 5, for 16 h, and live-imaged. Images were extracted from Supplementary Movie 1. **c–g** A549 cells were infected with PR8 WT virus, PA-GFP virus or transfected with a plasmid encoding GFP-NP or GFP and co-infected with PR8 virus, at an MOI of 5, for 16 h. **c** Cells were fixed, processed for FISH to detect segments 1 and 3, and imaged. In the upper left panel, cells were also stained for NP (green). At least 30 cells were analyzed per condition, from 2 independent experiments. **d–g** Cells were imaged under time-lapse conditions. Individual frames show fusion/fission events for PA-GFP virus (**d**, **f**) or GFP-NP/PR8 system (**e**, **g**) in the absence (**d**, **e**) or presence of the cytoskeleton drugs nocodazole and latrunculin A (**f**, **g**). Nocodazole and latrunculin A were added at 4 h. p.i. Yellow arrows highlight fusion or fission movements. Images were extracted from Supplementary Movies 2 to 9. Bar = 2 μm. At least 30 cells, from 3 independent experiments, were analyzed per condition

some viral inclusions might suffer liquid-to-solid transitions over the course of infection. In the GFP-NP/PR8 system, upon hexanediol treatment, we observed some cell blebbing . However, this has been described as a normal consequence of this treatment, and noted that it does not cause cell death, as cells recover upon washing out the drug[35].

Prolonged incubations with hexanediol are, nevertheless, toxic to the cells and lead to aberrant appearance of stress structures on account of molecular crowding promoted by cell shrinkage[35]. Therefore, we treated cells only for 30 min with 5% hexanediol before fixing them for quantifications, or before allowing cells to recover by washing out hexanediol and replacing with media. The methodology was the same as above, and results show that hexanediol treatment dissolved inclusions, although less efficiently than the hypotonic shock (from $88.1 \pm 5.0\%$ to $37.7 \pm 14.4\%$ for PR8; $82.0 \pm 32.8\%$ to $17.0 \pm 12.6\%$ for PA-GFP; $80.6 \pm 5.8\%$ to $30.3 \pm 15.9\%$ for GFP-NP/PR8, average percentage ± SD). Recovery from both treatments was equally effective ($92.6 \pm 2.9\%$ for PR8; $81.4 \pm 8.2\%$ for PA-GFP; $83.0 \pm 3.4\%$ for % for GFP-NP/PR8).

The ability of IAV inclusions to react to dilution and to hexanediol treatment suggests they have a liquid character[36]. Together, these data reveal that viral inclusions, containing both Rab11 and vRNPs, can respond to changes in the cellular environment and their constituents can self-organize into fluxional structures in live cells. In addition, the data convincingly show that cytosolic viral inclusions are similar for PR8, PA-GFP, and GFP-NP/PR8 infections, confirming that these systems can be used interchangeably.

**vRNP/Rab11 inclusions are dynamic membraneless organelles.** Our data indicate that viral inclusions are highly dynamic. To formally validate this notion, we used two strategies. First, we captured the movement of a 1 min movie in a snapshot, and showed the average intensity of labeled vRNPs (in red) as a defined puncta, surrounded by a wider green area that corresponds to the standard deviation of the average (Fig. 4a and Supplementary Movie 16). Second, we enquired if individual inclusions exchanged material with the exterior and performed Fluorescence Recovery After Photobleaching (FRAP). We found a high variation in the behavior of viral inclusions, with different speeds and patterns of recovery of the fluorescent signal. Some exhibited a fast and complete recovery (Fig. 4b, purple line, Supplementary Movie 16), while others showed a slower and incomplete recovery (Fig. 4b, blue line, Supplementary Movie 16). The recovery profile was also variable, with some regions losing and/or gaining intensity during the recovery phase (Fig. 4b, purple line), but others exhibiting a steady progression of fluorescence recovery (Fig. 4b, blue line). Not surprisingly, when the collection of FRAP events was averaged, the recovery profile obtained had a very large standard deviation (Fig. 4c, left graph). The calculated half time of recovery was 2.9 s and the diffusion rate calculated was $2.422 \pm 0.154 \, \mathrm{m}^{-13} \, \mathrm{s}^{-1}$ ($D \pm \mathrm{SEM}$), a value similar to what has been found for other liquid organelles

including Negri bodies formed during rabies virus infection[24]. These measurements are also consistent with those of nucleoli and stress granules[24], indicating that viral inclusions exchange material with similar structures or with the cytosol. The mobile fraction of vRNPs varied from $39.1 \pm 16.4\%$ (mean ± SD) at 5 s to $61.0 \pm 39.1\%$ within 60 s, and the curve plateaued after 15 s (Fig. 4c, right graph). The immobile GFP-NP must be biologically relevant. It either translates thermodynamically stabilized or kinetically trapped state of the protein, as in stable interactions (presumably among different vRNPs in each cluster), or a complex pattern of exchange between vRNPs and the exterior. These data are also consistent with a percentage of viral inclusions resistant to hexanediol treatment (Fig. 3c), and might indicate that some viral inclusions transit to a gel-like state.

Liquid organelles that react fast to stimuli lack a delimiting membrane[30]. Using electron microscopy, we previously showed that viral infection induced clustering of vesicles heterogeneous in size[14], which were recently renamed irregular coated vesicles (ICV)[17]. Areas of clustered vesicles matched viral inclusions by correlative light and electron microscopy (CLEM) and constitute, in high percentage, round-shaped molecular concentrates (quantified in Supplementary Fig. 3c). These concentrates are enriched in membranes at the core (Fig. 4d, black delimited arrowheads), but interestingly are not delimitated from the cytosol by membranes (Fig. 4d). Using double immunogold labeling (Fig. 4e), we confirmed the existence of electron-dense regions positive for NP (white arrowheads and small dots) as vRNP proxy, protruding from vesicles (black delimited arrowheads) positive for Rab11 (large dots). These clustered ICVs in membraneless viral inclusions were absent in mock-infected cells (Supplementary Fig. 3a and b). Collectively, these data suggest that vRNP/Rab11 inclusions are liquid membraneless organelles arising from phase separation.

**Viral inclusions form in the proximity of ER exit sites.** We next asked whether the assembly of IAV inclusions was spatially regulated. It was recently reported that vRNPs associate with the ER when leaving the nucleus and proposed that Rab11 would collect vRNPs from the ER for delivery to the surface[17]. Our electron microscopy data also show that the ER is constantly in close proximity to viral inclusions (Fig. 4e, black arrowheads). We, therefore, tested if inclusions are associated with the ER, by using antibodies against different ER markers or a cell line expressing a fluorescent tagged-ER membrane marker (HeLa Sec61β-Emerald)[37]. Confocal imaging of cells at different times post-infection failed to identify co-localization between the ER and viral inclusions (Supplementary Fig. 4a–c). However, from 8 hpi onwards, viral inclusions dispersed throughout the cytoplasm were frequently found juxtaposed ER tubules (Supplementary Fig 4a–c, inlets), suggesting an association between both structures. To gain insight into the dynamics of ER-viral inclusion association, live-cell imaging was performed. For this, HeLa Sec61β-Emerald cells were transfected with mCherry-NP and infected with PR8 (Fig. 5a and Supplementary Movie 17), or A549

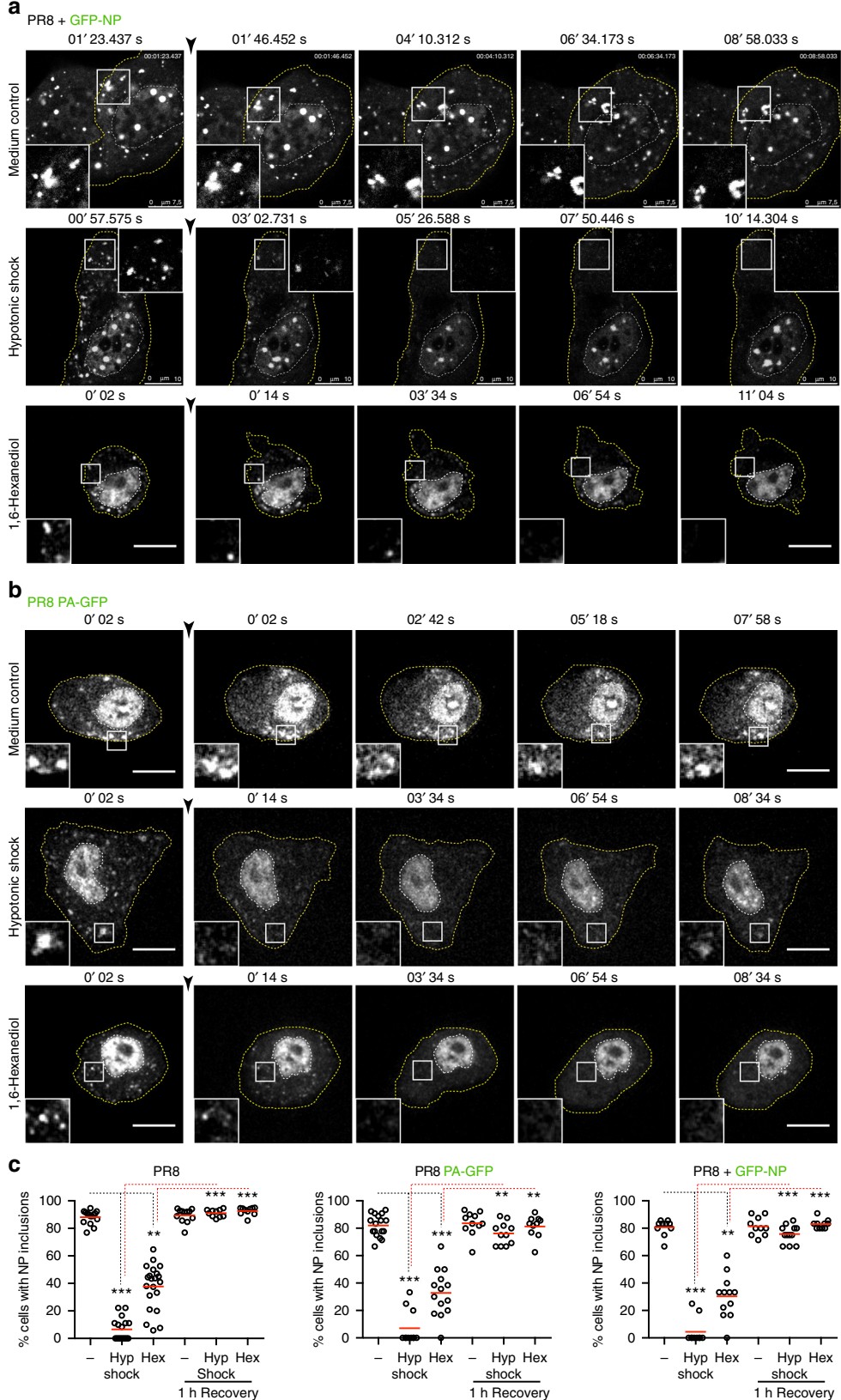

cells were co-transfected with mCherry-NP and ER-GFP and infected with PR8 virus (Fig. 5b and Supplementary Movie 18). In both experiments, viral inclusions displayed movements that matched those of the ER, although it is not clear whether ER motion was driving displacement of viral inclusions or,

conversely, if viral inclusions were gliding over the surface of ER tubules.

The ER is a complex organelle, with distinct morphologies and diverse functions[37,38]. In order to identify the specific ER domain interacting with viral inclusions, we tested different markers,

**Fig. 3** Viral inclusions quickly respond to changes in the cellular environment. A549 cells were transfected with a plasmid encoding GFP-NP and infected with PR8 (**a**) or infected with PA-GFP virus (**b**) at an MOI of 5. At 10–16 hpi, cells were imaged under time-lapse conditions. The black arrowhead indicates addition of 80% water (hypotonic shock), 5% hexanediol or regular growth medium. White boxes highlight viral inclusions in the cytoplasm in the individual frames. The dashed white line marks the cell nucleus, whereas the dashed yellow line delineates the cell periphery. Bar = 10 μm. Images were extracted from Supplementary Movies 10 to 15. Experiments were performed at least twice. **c** A549 cells were infected with PR8, with PA-GFP virus, or transfected with a plasmid encoding GFP-NP and infected with PR8 as above. At 16 hpi, cells were fixed and stained for NP. Percentage of cells with undissolved viral inclusions under the indicated treatments was calculated and plotted. Statistical analysis of data was performed using a non-parametric Kruskal–Wallis test, followed by Dunn's multiple comparisons test (**p < 0.05; ***p < 0.001). At least 10 cells were analyzed per individual dot and 10 panels per condition

including Atlastin 3, which accumulates in 3-way junctions, as well as Sec23 and Sec31A, both present in ER Exit Sites (ERES). We observed no correlation between Atlastin 3 and vRNPs staining (Supplementary Fig. 4d), but Sec23 and Sec31A localized frequently between the inclusions and the ER, and even co-localized with NP in specific spots (Fig. 5c, Supplementary Fig. 4e). Live-cell imaging of Sec16, another ERES component, indicated that these sites serve as docking platforms for vRNP accumulation. Here occurs a constant flux of material in and out of viral inclusions, as well as frequent fission and fusion events (Fig. 5d, Supplementary Movies 19 and 20).

**Viral inclusions depend on Golgi-ER vesicular cycling.** The above results suggest that the establishment of viral inclusions may also require the Golgi compartment. To address this issue, we inhibited the shuttling of cargo proteins between the Golgi and the ER by treating cells with brefeldin A (BFA)[39]. BFA, when added at 90 min post-infection to allow viral entry into cells, led to 1.5 log reduction in viral titres. The effect was more pronounced than that obtained for nocodazole, which resulted in a modest reduction of 0.6 log. Neither treatment interfered with NP expression (Fig. 6a, b), which suggests that BFA affects late stages of infection. Upon addition of a low dosage of BFA to cells infected for 8 h, viral inclusions disassembled within less than 1 h (Fig. 6b, lower panel). There was a significant decrease in the size [from 0.24 ± 0.20 μm² (mean ± SD) to 0.19 ± 0.15 μm²] and number of viral inclusions per μm² (from 0.25 ± 0.051 to 0.18 ± 0.056), as well as in the percentage of NP that was inside the inclusions (from 17.5 ± 4.8 to 9.2 ± 2.8%) (Fig. 6c). Immunostaining of cells with ER and Golgi makers (PDI and GM130, respectively) revealed that BFA treatment provoked the disassembly of the Golgi complex, but not of the ER (Fig. 6b). Areas stained for Rab11 also decreased with BFA treatment (Fig. 6d). Note that the viral transmembrane protein M2 still localized at the plasma membrane, likely because of low dosage (2 μg mL⁻¹) and short duration (1 h) of the BFA treatment (Fig. 6d)[39]. These results were confirmed by live-cell imaging using the GFP-NP/PR8 system. After 5 min of BFA addition, viral inclusions dissolved in contrast to untreated samples (Fig. 6e, Supplementary Movies 21 and 22). In sum, the data collectively show that biogenesis of IAV liquid inclusions enriched in vRNPs and Rab11 is dependent on continuous cycles of material between the ER and the Golgi, indicating that their distribution is spatially regulated.

The ERES are specialized domains where secretory proteins are loaded into coat protein complex II (COPII)-coated vesicles and transported to the Golgi[40]. Recruitment of COPII proteins to the ERES is controlled by the SAR1 GTPase cycle[41]. This small-GTPase also regulates ER membrane tubulation and vesicle fission, having a critical role in the generation of the ERES[42]. To analyze the effect of disrupting the ERES on the assembly of viral inclusions, we overexpressed a GTP-restricted mutant of GFP-tagged SAR1 (SAR1-GTP), which inhibits anterograde protein transport. Overexpression of GFP and GFP-tagged SAR1 WT

(SAR1) were performed as controls. Immunofluorescence analysis showed that overexpression of SAR1-GTP reduced the size of viral inclusions, when compared to overexpression of GFP or SAR1, in a statistically significant manner (Fig. 7a, b). The number of inclusions per μm² of cellular area and the percentage of NP signal present inside viral inclusions were also analyzed, with the latter being significantly reduced when SAR1-GTP was overexpressed [7.7 ± 2.9% (mean ± SD) in SAR1-GTP vs 12.7 ± 3.8% in GFP and 11.1 ± 3.4% in SAR1] (Fig. 7b). Confocal imaging of the viral transmembrane protein hemaglutinin (HA) confirmed that SAR1-GTP is disrupting ER-Golgi trafficking, since this protein was retained and accumulated in the ER when SAR1-GTP was overexpressed, but reached the plasma membrane during GFP and SAR1 overexpression (Fig. 7c).

**Viral inclusions do not promote escape to innate immunity.** It has been shown that phase-separated compartments are able to sequester or exclude specific material, including components of the innate antiviral immune response[43,44]. It is, therefore, possible that formation of IAV inclusions is a strategy to prevent the activation of cell-intrinsic defenses, either by sterically excluding sensors of exogenous material or by sequestering key factors of the downstream pathways. To address this hypothesis, we have used A549 cells constitutively expressing a fully functional or a nonfunctional form of GFP-tagged Rab11 [GFP-Rab11 wild type (WT) or GFP-Rab11 dominant negative (DN), respectively]. Both cell lines were infected with WT PR8 or an NS1 mutant virus that does not express a functional form of the main viral factor suppressing cell antiviral responses (NS1-N81)[45]. To characterize viral infection, cells were fixed at 8 and 16 hpi, stained for NP protein and imaged by confocal microscopy (Fig. 8a). Changes in Rab11 subcellular distribution were quantified by measuring the area of Rab11 inclusions in infected and control cells, and ranking them as above (Fig. 8b). As in our previous work[14], infection of cells stably expressing GFP-Rab11 WT with WT PR8 virus induced a redistribution of Rab11, forming large inclusions that contained vRNPs (Fig. 8b). Furthermore, the frequency of large Rab11 inclusions increased as infection progressed from 8 to 16 hpi. Noteworthy, infection of this cell line with the NS1 mutant virus produced similar changes in the frequency distribution of the different size category inclusions (Fig. 8b). Infection of GFP-Rab11 DN cell line, either with WT PR8 or NS1 mutant virus, did not change Rab11 DN distribution (Fig. 8a, b). Consistent with previous reports[13,18,46], Rab11 DN was primarily localized to the TGN, with some diffuse cytoplasmic staining also visible (Fig. 8a). Also in agreement with these studies, overexpression of Rab11 DN impaired the formation of viral inclusions characteristic of IAV infection[12,15], and therefore NP was diffusely distributed throughout the cytoplasm (Fig. 8a).

In order to investigate if impaired formation of viral clusters resulted in enhanced activation of the IFN cascade, the transcript levels of type I (IFN-α and IFN-β) and type III (IL-29) IFN, and of the IFN-stimulated gene viperin, were quantified at 8 and 16 hpi. For positive control, cells were transduced with the double-stranded

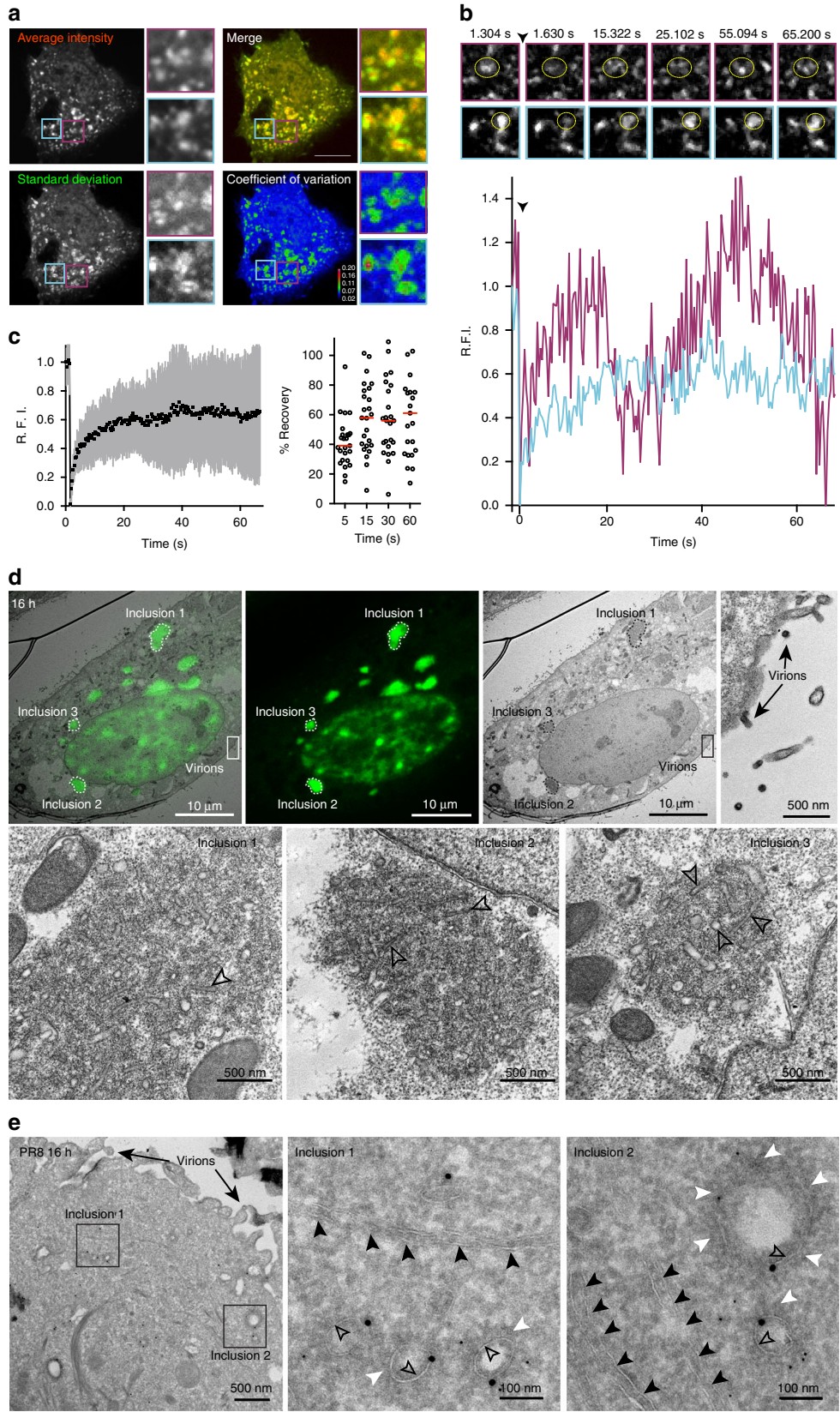

RNA mimic polyinosinic:polycytidylic acid [poly(I:C)]. Results show that there are no differences between both cell lines, in any of the conditions analyzed (Fig. 8c). Also, NS1 mutant virus induced higher mRNA levels than the WT virus, confirming the IFN-antagonizing role of NS1 (Fig. 8c). At the protein level, cell lysates from infected cultures were probed for active, phosphorylated IFN regulatory factor 3 (IRF3), a hallmark of activation of the IFN induction cascade (Fig. 8d), and cell culture media were tested for the levels of secreted IFN-β (Fig. 8e). Again, the results obtained were identical for both cell lines. In summary, all results point

**Fig. 4** vRNP/Rab11 inclusions exchange material dynamically and form membraneless liquid organelles. A549 cells were transfected with a plasmid encoding GFP-NP and infected with PR8 virus, at an MOI of 5 for 10–16 h, and imaged under time-lapse conditions. **a** A representative cell is shown. The fluorescence signal of viral inclusions in this cell is depicted as: average intensity (in red), standard deviation (in green), the merge of both, and coefficient of variation. Two areas of viral NP inclusions, highlighted in purple and cyan boxes, were selected for fluorescence recovery after photobleaching (FRAP). Bar = 10 μm. **b** The photobleached regions are marked by a yellow circle. The black arrowhead indicates the time of photobleaching. Relative Fluorescence Intensity (R.F.I.) was plotted as a function of time for each particle. Images have been extracted from Supplementary Movie 16. **c** R.F.I. was plotted as a function of time for the means of 25 FRAP events (left graph). The means are shown (black) with error bars representing the standard deviation (gray). The percentage of recovery of each photobleached region is shown for specific times (right graph), with medians represented as red bars. A single experiment representative of two independent experiments is shown. **d** HeLa cells with the GFP-NP/PR8 system, infected for 16 h (MOI = 10), were imaged by confocal and electron microscopy, and the resultant images were superimposed. Areas of correlation, inclusions 1 to 3, are delineated by a dashed line in the upper panel and shown in greater detail in the lower panel. Progeny virions budding at the surface (black arrows) show that the cell is infected. Black delimited arrowheads show individual vesicles within the inclusion. **e** GFP-Rab11 WT cells were infected with PR8 virus (MOI = 5) for 16 h, and then stained for GFP (18 nm gold particles) and viral NP (6 nm gold particles). Inclusion areas are highlighted by black boxes. Black arrowheads indicate ER structures in the vicinity of viral inclusions. Black arrows show progeny virions budding at the cell surface. Black delimited arrowheads show vesicles. White arrowheads show electron-dense vRNPs. A single experiment representative of two independent experiments is shown

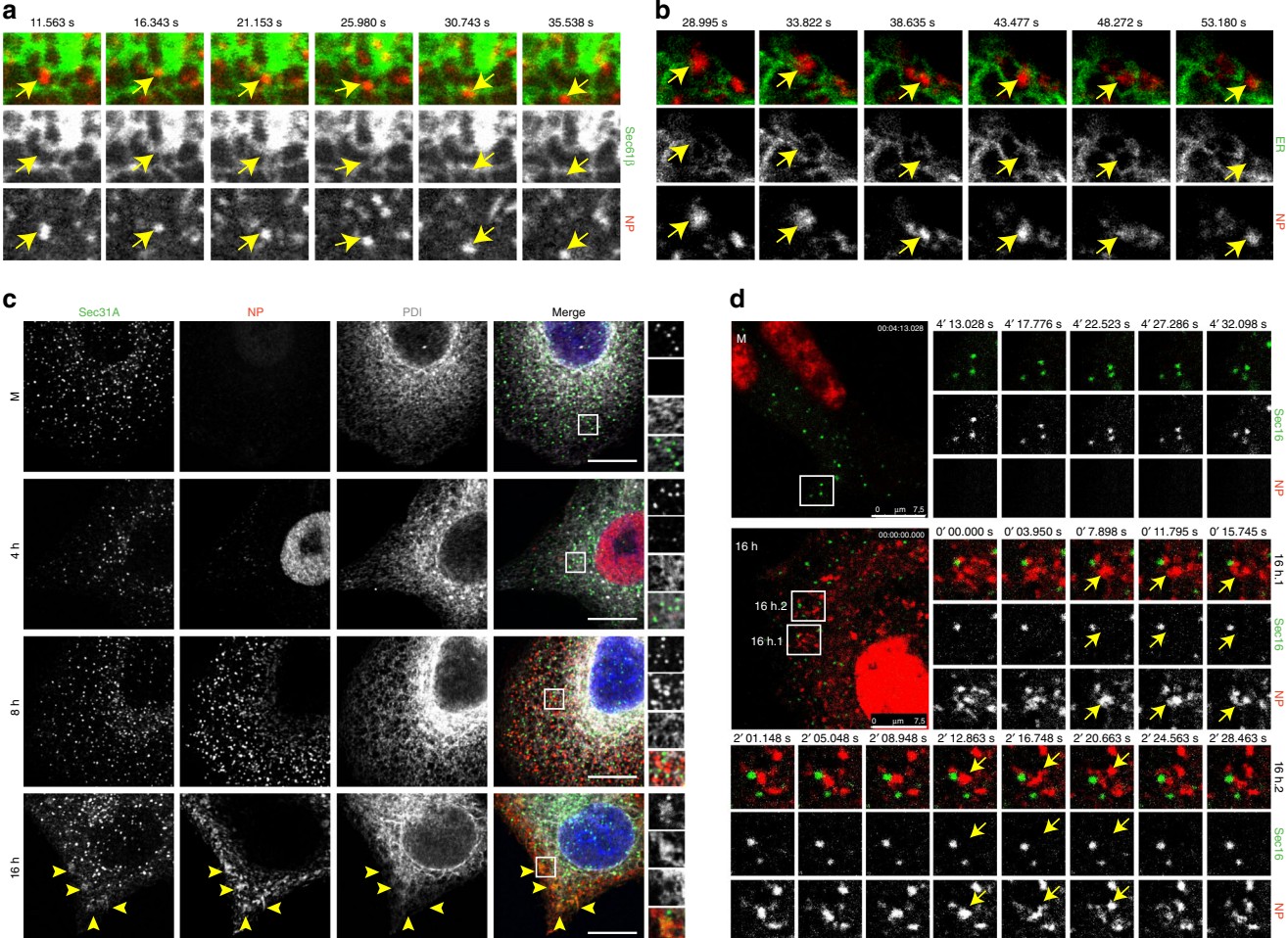

**Fig. 5** Viral inclusions are associated with ER exit sites. **a** Sec61β-Emerald cells were transfected with mCherry-NP and infected with PR8 virus, at an MOI of 10, for 16 h. **b** A549 cells were co-transfected with plasmids encoding mCherry-NP and ER-GFP and infected with PR8 virus, at an MOI of 10, for 16 h. **a**, **b** Cells were imaged under time-lapse conditions. Individual frames with single moving particles highlighted with yellow arrows are shown. Images were extracted from Supplementary Movies 17 and 18. More than 15 cells from 3 independent experiments were analyzed in each condition. **c** A549 cells were infected or mock-infected (M) with PR8 virus, at an MOI of 3, and fixed at the indicated times. Cells were stained for the ER proteins Sec31A (in green) and PDI (in gray) and the viral NP protein (in red). Areas highlighted by the white box are shown on the right of each panel. Yellow arrowheads indicate co-localization between Sec31A and NP. Bar = 10 μm. More than 30 cells from 2 independent experiments were analyzed. **d** A549 cells were co-transfected with plasmids encoding mCherry-NP and GFP-Sec16 and infected or mock-infected (M) with PR8 virus for 16 h. Cells were imaged under time-lapse conditions. Representative cells are shown in the left large images. Individual frames with single moving particles highlighted with yellow arrows are shown in the small panels. Two examples are provided for the infected cell (16 h.1 and 16 h.2). Bar = 7.5 μm. Images were extracted from Supplementary Movies 19 and 20. Images are representative of at least 15 cells, from 2 independent experiments

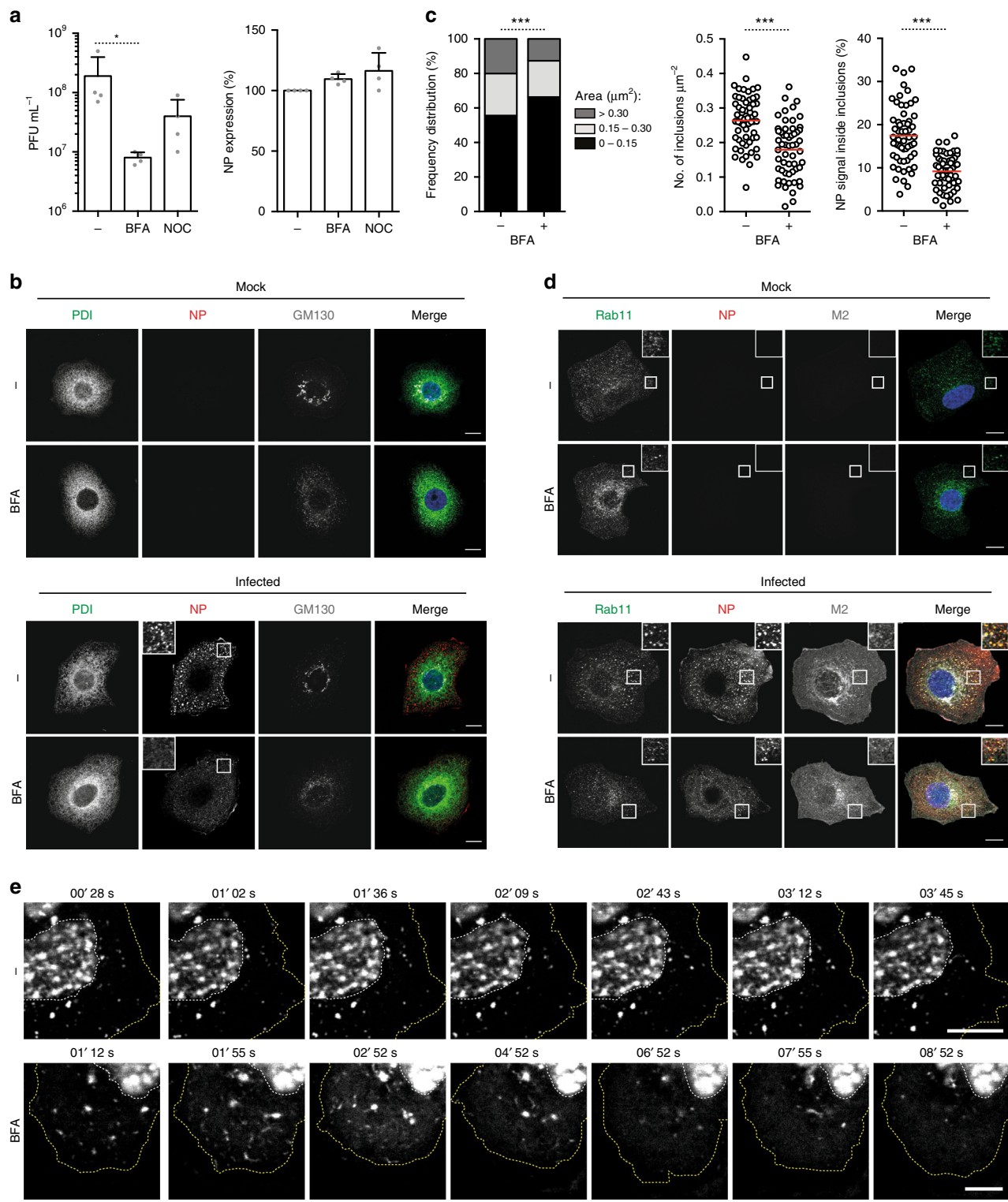

towards innate immune responses not being affected by biological phase transitions of vRNPs.

Next, we examined the impact of constitutively expressing GFP-Rab11 WT or DN on viral replication, by plaque assay (Fig. 8f). NS1-N81 virus production was, as expected[47], attenuated in both cell lines, as compared to WT PR8 virus (ranging from 0.4 log at 16 hpi for growth in Rab11 WT lines to 0.9 log for the same lines at 8 hpi, Fig. 8f). When compared with viral growth in GFP-Rab11 WT, production of NS1-N81 and WT viruses in GFP-Rab11 DN

cell line were reduced 0.4–0.8 log, values lower than previously reported in refs. [13,46] (Fig. 8f).

To investigate these discrepancies, we decided to sort the lines for low and high expression of GFP-tagged proteins (Supplementary Fig. 5a) and evaluated viral production. Results showed that cells expressing high and low levels of GFP, or low levels of GFP-Rab11 WT were equally permissive to viral infection at 8, 10, and 16 hpi (Supplementary Fig. 5b). Conversely, high expression levels of GFP-Rab11 WT originated

**Fig. 6** Disruption of ER-Golgi trafficking disassembles vRNP hotspots. A549 cells were infected or mock-infected with PR8 virus at an MOI of 3. **a** At 90 min p.i., 2 μg mL$^{-1}$ of brefeldin A (BFA) or 10 μg mL$^{-1}$ nocodazole (NOC) were added and incubated for 10 h until the supernatant was collected and viral titres determined or cells were harvested in Laemmli's and NP levels detected by western blotting. Statistical analysis of data was performed using a non-parametric one-way ANOVA, followed by Friedman's multiple comparisons test (*$p < 0.05$). **b–d** At 8 hpi, cells were also treated or mock-treated with 2 μg mL$^{-1}$ of BFA for 1 h. **b** Cells were immunostained for the ER marker PDI (in green), the viral protein NP (in red) and the *cis*-Golgi marker GM130 (in gray) and imaged by confocal microscopy. Selected areas of the cytoplasm are marked by white boxes and displayed on the top left corner of the images. Bar = 10 μm. **c** The frequency distribution of NP inclusions within the three size categories (in μm$^2$), the number of inclusions per μm$^2$, and the percentage of NP staining that is inside inclusions were plotted for each condition. Statistical analysis of data was performed using a non-parametric Kruskal–Wallis test, followed by Dunn's multiple comparisons test (***$p < 0.001$). An average of 60 cells from 2 independent experiments was analyzed per condition. **d** Infected cells were stained for the host protein Rab11 (in green) and the viral proteins NP (in red) and M2 (in gray). Cells were imaged by confocal microscopy. Areas highlighted by the white box are shown on the right top corner of each image. **e** A549 cells were transfected with a plasmid encoding GFP-NP and co-infected with PR8 virus, at an MOI of 5, for 10 h. Cells were imaged under time-lapse conditions in the absence or immediately after adding 2 μg mL$^{-1}$ of BFA. Bar = 10 μm. Images were extracted from Supplementary Movies 20 and 21 , and are representative of 9 videos from 2 independent experiments

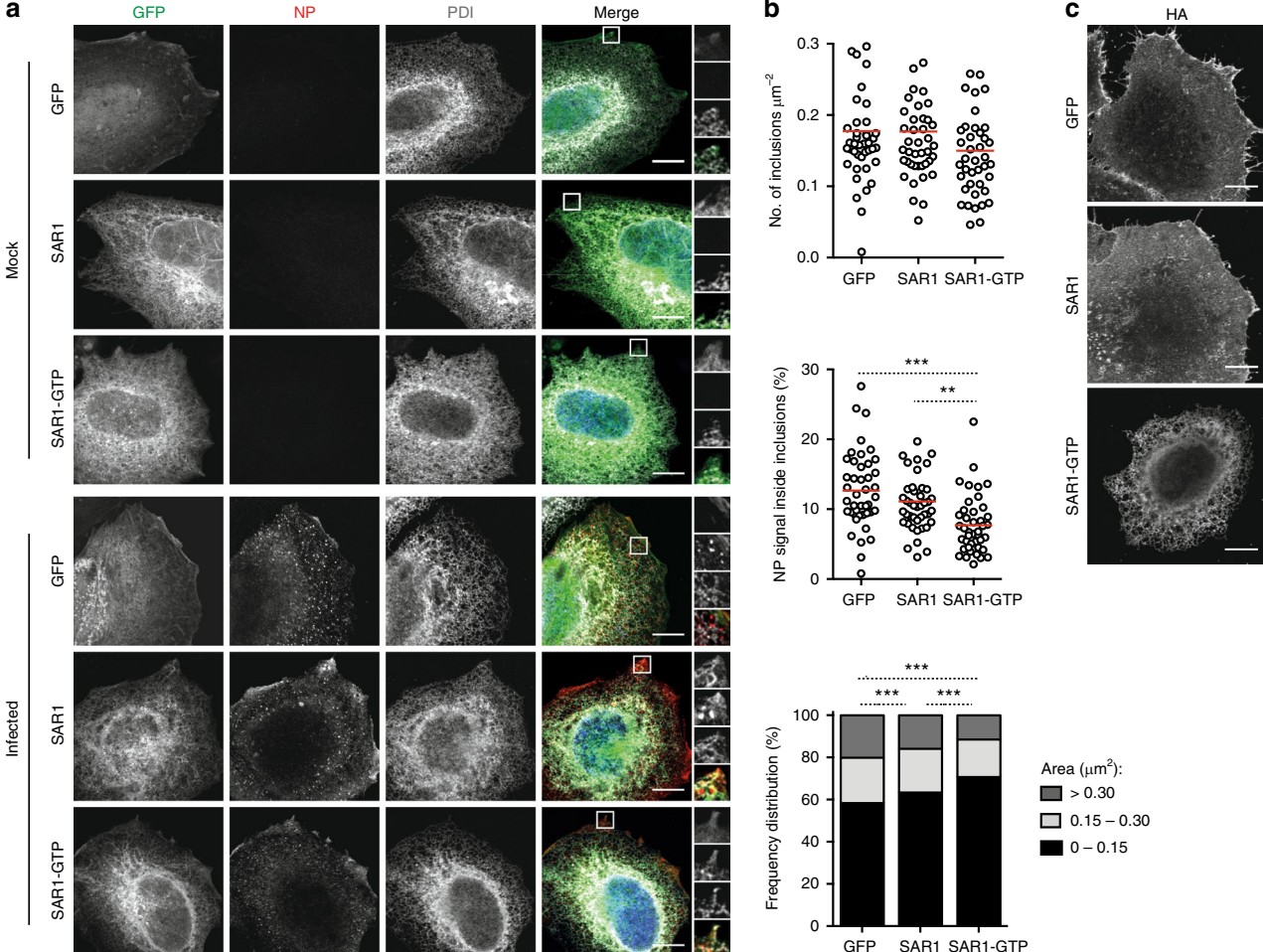

**Fig. 7** Disruption of ER exit sites dissolves viral inclusions. HeLa cells were transfected with plasmids encoding GFP, SAR1 WT-GFP (SAR1) or SAR1-GTP-GFP (SAR1-GTP) and, 24 h later, infected or mock-infected with PR8 virus, at an MOI of 10. At 16 hpi, cells were fixed and processed for immunofluorescence. **a** Cells were stained for the viral protein NP (in red) and for the ER protein PDI (in gray). Areas highlighted by the white box are shown on the right of each panel. **b** The number of inclusions per μm$^2$, the percentage of NP staining that is inside inclusions and the frequency distribution of NP inclusions within the three area categories (in μm$^2$) were plotted for each condition. Statistical analysis of data was performed using a non-parametric Kruskal–Wallis test, followed by Dunn's multiple comparisons test (***$p < 0.001$). More than 40 cells from 2 independent experiments were analyzed per condition. **c** Infected cells were stained for the viral protein HA and imaged by confocal microscopy. Bar = 10 μm

a significant decrease in viral titres of over 1 log for all time points. Curiously, this decrease was similar to the observed for low levels of GFP-Rab11 DN and the value decreases again by over 3 logs when GFP-Rab11 DN levels were high (Supplementary Fig. 5b). Therefore, GFP-Rab11 DN is the factor with higher impact on viral infection, and efficiently abrogates viral production, as observed before[13,46]. However, the data also indicate that above an optimal range, even Rab11 WT has a

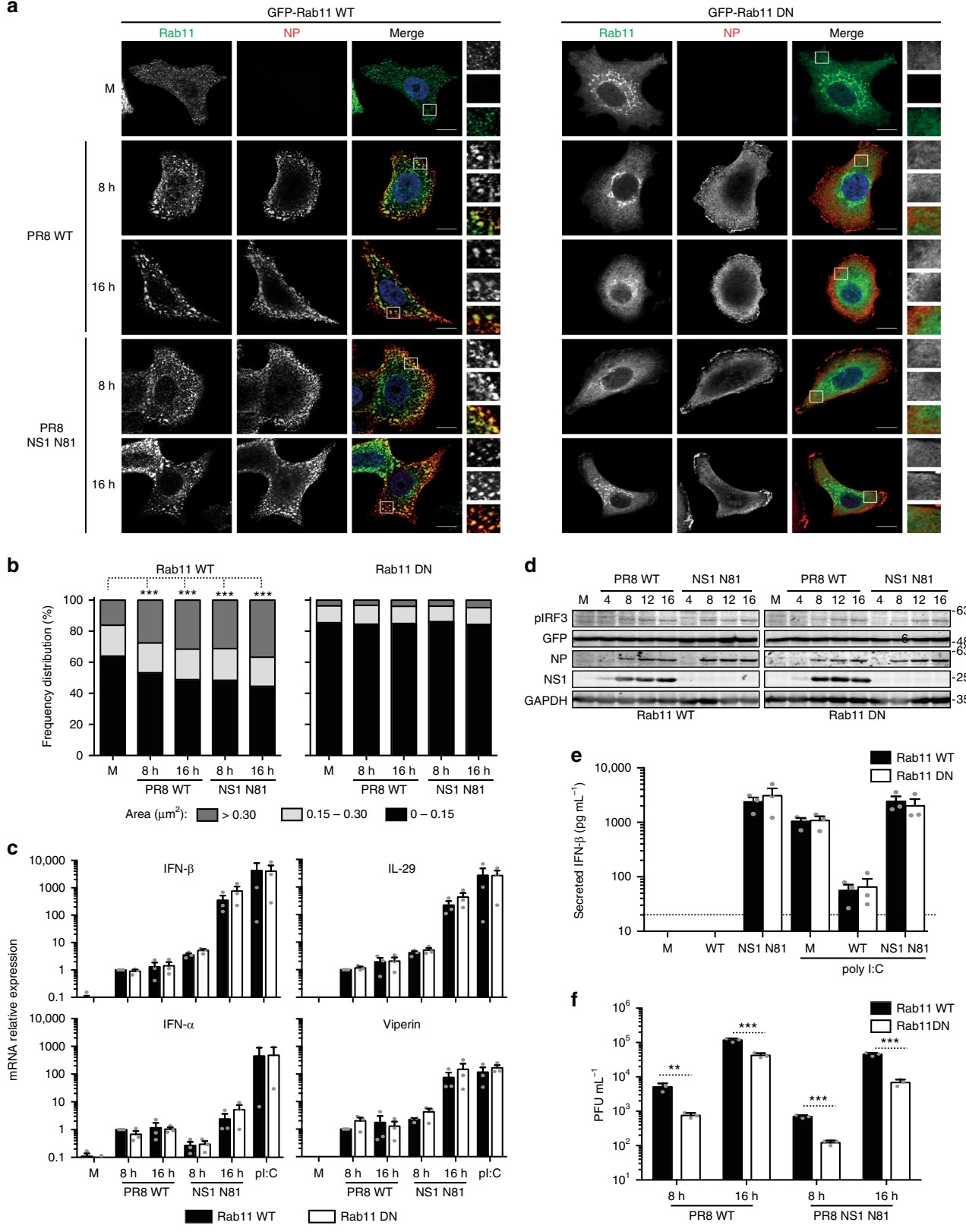

detrimental effect in viral replication. Future work should investigate if this is related with viral inclusions.

**Viral inclusions host vRNPs of different parental viruses**. The observation that expression of a single vRNP could drive

formation of viral inclusions has led us to hypothesize that these structures could operate as dedicated spots for IAV genome complex formation. This new model, consistent with a selective process for assembling the supra-molecular genomic complex, would restrict vRNPs in space and time and increase kinetics

**Fig. 8** Interferon response is not affected by the formation of viral inclusions. **a** Stable cell lines expressing GFP-Rab11 WT or GFP-Rab11 DN were infected or mock-infected (M), at an MOI of 3, with PR8 WT or NS1-N81 viruses. **a** Cells were fixed at 8 and 16 hpi and stained for NP (in red). Bar = 10 μm. **b** The frequency distribution of NP inclusions within the three area categories (in μm$^2$) was plotted for each cell line. Statistical analysis of data was performed using a non-parametric Kruskal–Wallis test, followed by Dunn's multiple comparisons test (*** p < 0.001 for GFP-Rab11 WT cells; no statistical significance found for GFP-Rab11 DN cells). Statistical analysis compares the area of all inclusions between conditions. An average of 30 cells was analyzed per condition. A single experiment representative of two independent experiments is shown. **c** Expression of IFN-β, IFN-α, IL-29 and viperin was evaluated at the level of transcription by RT-qPCR in relation to GAPDH. Poly(I:C) was used as a positive control for maximum expression of these transcripts. Statistical analysis of data was performed using two-way ANOVA test, followed by Sidak multiple comparisons test (no statistical significance between conditions found). Data represents the average of three independent experiments. **d** Expression of phosphorylated IRF3, GFP, NP, NS1, and GAPDH was evaluated at the protein level by western blotting. **e** The levels of secreted IFN-β were quantified by ELISA in cell supernatants at 24 hpi. Poly(I:C) was used as a positive control for maximum expression of IFN-β protein. The limit of detection of this method is 30 pg mL$^{-1}$ (dashed line). Statistical analysis of data was performed using two-way ANOVA test, followed by Sidak multiple comparisons test (no statistical significance between conditions found). Data represent the average of three independent experiments. **f** At the indicated times, supernatants were collected and viral production was evaluated by plaque assays using MDCK cells. Statistical analysis of data was performed using Holm–Sidak multiple comparisons test (**p < 0.01, ***p < 0.001). Data correspond to one representative experiment out of three independent experiments

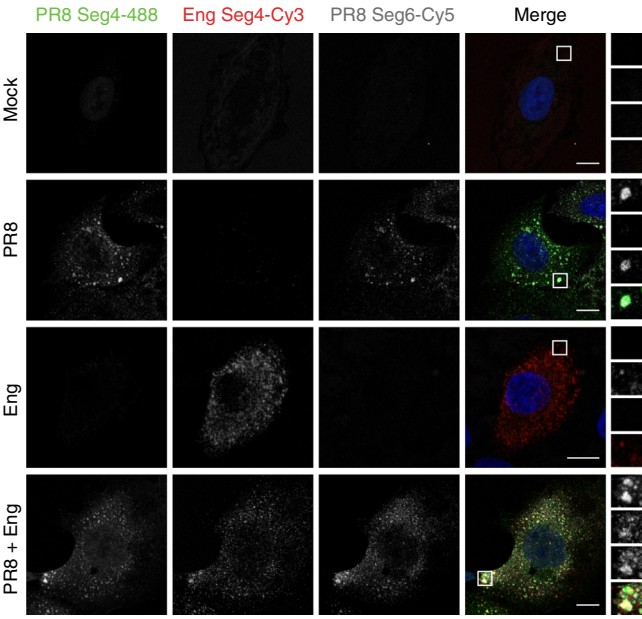

**Fig. 9** Viral inclusions harbor vRNPs of two parental viruses in co-infections. A549 cells were mock infected or infected with PR8 and/or Eng2009, at an MOI of 3, each. At 16 hpi, cells were fixed and processed for FISH to detect segments 4 of both viruses and segment 6 of PR8. Inlets show magnifications with all segments colocalizing in co-infections, and segment 4 and 6 upon PR8 challenge. 10 cells were analyzed per condition

of biochemical reactions compared to vRNPs scattered throughout the cytosol. We reasoned that if this was the case, vRNPs from two parental strains would collocate in viral inclusions. To test this idea, A549 cells infected or mock-infected with PR8 and/or influenza A/England/195/2009 (Eng2009) were analyzed by FISH at 16 hpi. Mock-infected cells did not highlight segments 4 or 6 of both viruses (Fig. 9, top panels). Infection with PR8 positively marked segment 4 and 6 of PR8 in cytosolic viral inclusions, but not segment 4 of Eng2009. Conversely, Eng2009 selectively detected segment 4 of this virus in cytosolic puncta, showing that the probes used are specific, distinguishing between Eng2009 and PR8. In co-infection, viral inclusions hosted segment 4 of both viruses as well as segment 6 of PR8. Collectively, the data show that viral inclusions could serve as sites for IAV genome assembly, including those of reassortant viruses (Fig. 9).

## Discussion

Viral inclusions that emerge during IAV infection have been linked to genome assembly. In this work, we provide evidence that viral inclusions are formed when cells express a single vRNP type (Fig. 1). Given that vRNPs of the same type compete for inclusion in virions[8–10,48], our findings indicate that formation of these structures precedes vRNP–vRNP interactions[8–10,48]. This leads to a paradigmatic change in the model of IAV genome assembly. We propose that viral inclusions allow the spatio-temporal control of the genome assembly process. Such mechanism would require concentrating material in the cytosol, with a constant influx of vRNPs and efflux of assembled genomes (Fig. 10).

Supporting this premise, mechanisms to isolate selected molecules from the cytosol[31] without the need of a membrane were recently described[49]. Such compartmentalization, driven by liquid demixing, originates functional phase-separated organelles of components and reactions. Several membraneless organelles in the cell (nucleolus, centrosomes, stress granules, DNA repair foci, or G bodies) respond and adapt quickly to stimuli[30,49]. Interestingly, viral-induced membraneless territories have been known for decades and functionally associated with host immune escape, viral replication, and assembly[25,43]. Despite the similarities, the two biological assemblages have been treated as unrelated phenomena and the link between them is still missing.

Herein, we report that IAV forms viral inclusions with liquid-like properties (Figs. 1 and 2). Although they share physical characteristics with Negri bodies found in rabies virus-infected cells in terms of shape, dynamism, and ability to deform[44], they are not involved in viral replication, which takes place in the host cell nucleus. In addition, they behave as other bodies formed by liquid phase separation, including reacting fast to physiological changes (Fig. 3)[36,50]. Formation of these inclusions during IAV infection depends on Rab11-GTP and vRNPs. Despite shared characteristics between the molecules involved in the formation of IAV inclusions and other membraneless bodies—such as multivalency (Rab11)[51,52], internally disordered regions (NP)[36,53], nucleic acids (vRNPs)[54], and oligomerizing RNA binding proteins (NP)[55]—the rules underlying their biogenesis and function in infection are far from understood. For example, it is unclear which key constituent(s) can phase separate, capture vRNPs, and cluster vesicles and their attached components[56].

Interestingly, cross-talks between classical and liquid organelles are starting to emerge[57]. In this manuscript, we also provide a link between the two systems, demonstrating that formation of liquid viral inclusions is spatially regulated, developing near the

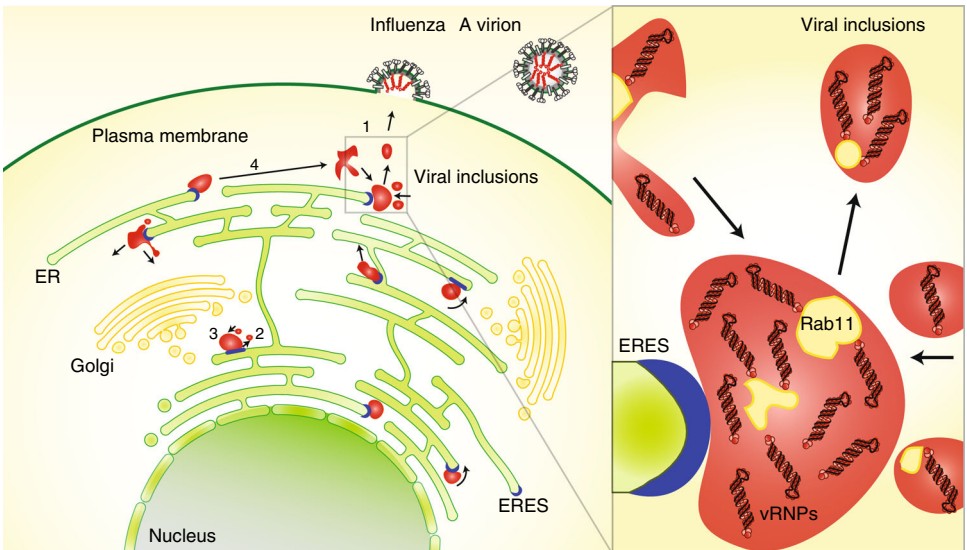

**Fig. 10** Proposed model. Viral inclusions (in red) have properties of liquid organelles, segregating from the cytosol without a delimiting membrane. Viral inclusions exchange material dynamically (1) and deform easily, exhibiting fission (2) and fusion (3) events. Viral inclusions can travel long distances before and after fusion/fission events (4), respectively. These organelles are formed in the vicinity of ERES (in blue) and their assembly is dependent on continuous ER-Golgi vesicular cycling. We propose that viral inclusions trigger nucleation of vRNP–vRNP interactions among the eight different segments to assemble a complete IAV genome. Inlet shows composition of viral inclusions close to ERES. These contain vRNPs of all types, Rab11 and host membranes are clustered, but not delimited by a lipid bilayer

ER (Fig. 10). In agreement, we observe that movement of IAV inclusions matches that of the ER (Fig. 5) and, in some cases, inclusions seem to slide on or with ER membranes (Supplementary Movies 17 and 18). This raises the possibility that viral inclusions move using the ER. This hypothesis, together with the finding that vRNPs attach the ER[17], might help answering the unresolved question of vRNP transport to the plasma membrane. Discrepant studies found vRNPs moving with speeds consistent with using microtubules, but nocodazole unexpectedly only had a mild inhibitory effect on viral production[11,58,59]. However, ER sliding, which uses microtubules, is resistant to this drug[60]. In this study, nocodazole did not hinder viral inclusion formation (Fig. 2e, f), agreeing with vRNP movement on the ER, but severely impaired fusion and fission events (Fig. 2e, f)[17,60,61]. Interestingly, a recent study found that nocodazole prevented the association between Rab11 and vRNPs[61]. Whether Rab11 is involved in material exchange among inclusions remains to be evaluated. Compelling evidence to validate a model in which upon nuclear export, vRNPs move using the ER and concentrate near the ERES to assemble viral genomes for posterior delivery to the plasma membrane by Rab11 vesicles is still needed. Unresolved questions include: (1) routing of Rab11 to the ER; (2) binding of vRNPs to the ER; (3) selection of the ERES as scaffolds for viral inclusions, (4) formation of a supra-molecular complex inside these structures and in the case that occurs, (5) sensing and transport of fully assembled genomes to the plasma membrane.

Despite lack of mechanistic insight, formation and maintenance of viral inclusions is likely a conserved and robust process. Co-infection with two distinct human IAV strains resulted in the co-lodging of their vRNPs in the same inclusions (Fig. 9). Formation of viruses with mixed genomes from human and avian adapted strains (reassortant viruses) has been associated with pandemic viruses and excess mortality. Hence, development of viral inclusions should be analyzed in more detail to unravel their contribution to production of reassortant strains.

It is not surprising that liquid bodies could be involved in IAV RNA processes, since an intimate association between RNA and liquid membraneless organelles has been consistently reported[62–65]. In some cases, RNA promotes phase separation[62,63], while in other it inhibits this process[64,65]. Phase separation could also play other roles during IAV infection, such as escape from host immune detection. However, we found no differences in the activation of IFN response when assembly of viral inclusions was inhibited (Fig. 8). Nevertheless, other immune parameters need to be examined before discarding a role of inclusions in protecting against immune recognition. Finally, as shown for other systems, phase separation could operate in signal amplification[66] or repression/activation of specific cellular pathways by exclusion/inclusion of sets of molecules[67].

In summary, we propose that the role of liquid viral inclusions is to spatially restrict vRNPs, increasing their concentration at specific sites to boost the kinetics of genome assembly. Future experiments should detail the internal organization of the viral inclusions and factors affecting liquid properties, such as molecular crowding, solubility, affinity, or the valency of phase-separating proteins[67]. In addition, identifying the key features of viral inclusions could provide means to control their formation and maintenance during IAV infection. We are just beginning to understand the involvement of phase separation in virology, but we anticipate that, given the ancient co-evolution between viruses and eukaryotic cells and the diversity of host strategies used by viruses, the next years will provide an interesting overlap between the two fields.

## Methods

**Cells, Viruses and Drugs**. The epithelial cells Madin-Darby Canine Kidney (MDCK), Human Embryonic Kidney 293 T (HEK293T), human cervical HeLa and human alveolar basal (A549) were a kind gift of Prof Paul Digard, Roslin Institute, UK. The GFP-Rab11 WT and GFP-Rab11 DN cell lines (A549) were produced by our laboratory[14]. The Sec61β-Emerald (HeLa) cell line was a kind gift from Dr. Christoph Dehio, Biozentrum, University of Basel, Switzerland[37,68]. All cell types were cultured in Dulbecco's Modified Eagle's medium (DMEM, Gibco, 21969035) supplemented with 10 % fetal bovine serum (FBS, Gibco, 10500064), 1% penicillin/

streptomycin solution (Biowest, L0022) and 2 mM L-glutamine (Gibco). GFP-Rab11 (A549) and Sec61β-Emerald (HeLa) culture media was also supplemented with 1.25 µg mL$^{-1}$ Puromycin (Calbiochem). Cells were regularly tested for mycoplasma contamination with the LookOut mycoplasma PCR detection kit (Sigma), using JumpStart Taq DNA Polymerase (Sigma). Reverse-genetics derived A/Puerto Rico/8/34 (PR8 WT; H1N1) was used as a model virus and titrated by plaque assay in MDCK cells[14]. NS1-N81 mutant virus was derived from PR8 WT and expresses only the first 81 amino acids of NS1[45]. PR8 PA-GFP virus was constructed as in Bhagwat et al., using the plasmids described below[33]. PR8 WT, PR8 NS1-N81, and PR8 PA-GFP viruses were rescued in HEK293T cells using Lipofectamine 2000 (ThermoFisher) and amplified in MDCKs or embryonated eggs. The circulating virus influenza A/England/195/2009 (Eng2009) was also used in this work (kind gift from Prof. Paul Digard). For viral growth curves, A549 cells were seeded in poly-d-lysine coated wells and, 16 h later, infected at a multiplicity of infection (MOI) of 0.001 in DMEM supplemented with 2 mM L-glutamine, 1% penicillin/streptomycin for 1 h at 37 °C and 5% CO$_2$. Virus were then removed and cells were grown in DMEM supplemented with 2 mM L-glutamine, 1% penicillin/streptomycin, 1 µg mL$^{-1}$ TPCK-trypsin (PAA) and 0.14% BSA (Sigma) until the supernatants were collected at different time points. The supernatants were subjected to a plaque assay to calculate the virus titres. All other virus infections were performed at an MOI of 3–10. After 45 min, cells were overlaid with complete culture media. The drug brefeldin A (Sigma) was dissolved in ethanol and used at final concentration of 2 µg mL$^{-1}$. Nocodazole (Sigma) and latrunculin A (Sigma) were dissolved in DMSO and used at a final concentration of 10 µg mL$^{-1}$ and 1 µM, respectively.

**Plasmids.** Reverse genetic plasmids were contributed by Dr. Ron Fouchier, Erasmus MC, Netherlands. HA-tagged Sec23 plasmid was a kind gift from Dr. Colin Adrain, IGC, Portugal. GFP-tagged Sec16 plasmid was purchased from Addgene. GFP-Sec61β was constructed by PCR-amplifying Sec61β from A549 cDNA and cloning it into pEGFP-C2, using HindIII and KpnI restriction sites. SAR1A was amplified from A549 cDNA and cloned into XhoI-BamHI restriction sites of pEGFP-N1. GTP-restricted SAR1 (H79G) was produced by site-directed mutagenesis (SDM) from SAR1 WT-GFP. ER-GFP plasmid was made from pEGFP-C2, by inserting a C-terminal KDEL sequence by SDM, and an N-terminal ER-signal sequence from calreticulin, by oligo-annealing between NheI and AgeI restriction sites. Plasmids used for the mini-replicon system have been described in reference[11], except pcDNA3-NS2. The latter was made by PCR amplification of NS2 (from PR8) and insertion in pCDNA3, using EcoRI and NotI restriction sites. pPol-I-luc (with a human RNA pol I promoter) has been described elsewhere[69]. For PA-GFP virus construction, a plasmid encoding GFP was sub-cloned into pCDNA3 using NotI and XbaI restriction sites subsequently from being amplified from pEGFP-C2. Plasmid encoding PA-GFP was sub-cloned into pCDNA3 GFP using EcoRI and NotI restriction sites subsequently from being amplified from PR8 pDual Seg3. A silent XbaI site was introduced in PA both in pCDNA3 PA-GFP and PR8 pDual Seg3 by SDM. The XbaI-digested fragment of PA-GFP was then inserted into PR8 pDual Seg3. pCDNA3 plasmids used to synthesize fluorescent in situ hybridization (FISH) probes to detect vRNA from segments 1, 3, 7, and 8 are described elsewhere[11]. Plasmids used to synthesize FISH probes to detect vRNA from PR8 segments 4 and 6, and from Eng2009 segment 4 in co-infection experiments were constructed by PCR-amplifying nt 533 to 910 of PR8 HA sequence, nt 55 to 181 of PR8 NA sequence, and nt 531 to 871 of Eng2009 HA sequence, respectively, and cloning them into pCDNA3, using HindIII and XbaI restriction sites.

The following primers/oligos were used:
Sec61β Fw: 5'-TAGAAAGCTTCATGCCTGGTCCGACCC-3'
Sec61β Rv: 5'-TCGAGGTACCCTACGAACGAGTGTACTTGCCC-3'
SAR1 WT Fw: 5'- TCGACTCGAGATGTCTTTCATCTTTGAGTGGATCT- 3'
SAR1 WT Rv: 5'- TCGAGGATCCCGGTCAATATACTGGGAGAGCCAGC- 3'
SAR1 H79G FW: 5'- TTTTGATCTTGGTGGGGGCGAGCAAGCACGTCGC - 3'
SAR1 H79G RV: 5'- GCGACGTGCTTGCTCGCCCCCACCAAGATCAAAA - 3'
KDEL Fw: 5'-TGGACGAGCTGTACAAGGACGAGCTGTAATCCGGCCGGACT-3'
KDEL Rv: 5'- AGTCCGGCCGGATTACAGCTCGTCCTTGTACAGCTCGTCCA-3'
Calreticulin tag up: 5'- CTAGCATGCTGCTATCCGTGCCGTTGCTGCTCGGCCTCCTCGGCCTGGCCGTCGCA-3'
Calreticulin tag down: 5'- CCGGTGCGACGGCCAGGCCGAGGAGGCCGAGCAGCAACGGCACGGATAGCAGCATG-3'
NS2 Fw: 5'-CGTAGCGAATTCATGGATCCAAACACTG-3'
NS2 Rv: 5'-GCTAAGACGCGGCCGCTTAAATAAGCTGAAAC-3'
GFP NotI Fw: 5'-GTCAGAATGCGGCCGCcATGGTGAGCAAGGGCGAGGAG-3'
GFP XbaI Rv: 5'-TCAGTCTAGATTACTTGTACAGCTCGTCCATGC-3'
PA pCDNA3 EcoRI_Fw: 5'-CGACGAATTCATGGAAGATTTTGTGCG-3'
PA pCDNA3 NotI Linker Rv: 5'-GTCGTCGCGGCCGCCACTCAATGCATGTGTAAGGAATGAG-3'
pDual Seg3 SDM Fw: 5'-CTAGAAGGATTTTCAGCTGAATCTAGAAAACTGCTTCTTATCGTTC-3'
pDual Seg3 SDM Rv: 5'-GAACGATAAGAAGCAGTTTTCTAGATTCAGCTGAAAATCCTTCTAG-3'

PR8 HA probe Fw: 5'-GCGTAAGCTTGACGGAGAAGGAGGGCTCAT-3'
PR8 HA probe Rv: 5'-GCGTTCTAGAGTGTTACACTCATGCATTGATGCG-3'
PR8 NA probe Fw: 5'-GCGTAAGCTTCAATCTGTCTGGTAGTCGGA-3'
PR8 NA probe Rv: 5'-GCGTTCTAGACCGGTTAATATCACTGAAGTTG-3'
Eng HA probe Fw: 5'-GCGTAAGCTTGCTCAGCAAATCCTACATTAATG-3'
Eng HA probe Rv: 5'-GCGTTCTAGACGTGGACTGGTGTATCTGAA-3'

**Transfections.** Cells, grown to 70% confluency in 24-well plates, were transfected with 250 ng of indicated plasmids or 100 ng of the synthetic dsRNA polyinosinic:polycytidylic acid [poly(I:C); Calbiochem], using Lipofectamine LTX (Life Technologies) and Opti-MEM (Life Technologies), according to manufacturer's instructions. Cells were infected or mock-infected 16 h post-transfection or simultaneously with transfection (live-cell imaging) at indicated MOI.

To reconstitute GFP-tagged vRNPs, 293 T cells grown to 70% confluency in 24-well plates were transfected with plasmids pcDNA PB1, PB2, PA (130 ng each), NP (150 ng), GFP-NP (50 ng), pPol-I segments 7 and 8 (130 ng each), or/and pcDNA-NS2, using Lipofectamine 2000 (Life Technologies) according to the manufacturer's instructions, incubated overnight, and imaged 12 to 16 h later.

**Confocal fixed-cell imaging.** For FISH analysis, cells were fixed for 20 min using 4% formaldehyde in PBS, washed three times in PBS and permeabilized using 0.2% (v/v) Triton-X-100 in PBS for 7 min followed by more PBS washes. Cells were postfixed using a 1%(v/v) formaldehyde solution in PBS for 10 min and washed again with PBS. The prehybridization mix (60% formamide, 0.3 M sodium chloride, 30 mM sodium citrate pH 7.0, 10 mM EDTA pH 8, 35 mM dextran sulphate (w/v), 250 ng mL$^{-1}$ tRNA) was added to the cells and incubated for 1 h at 37 °C. During this period, 488-, Cy3-, or Cy5-labbeled ribonucleotide probes were boiled for 5 min and placed on ice for further 5 min prior to dilution to a final concentration of 0.5% (v/v) in 0.3 mL of prehybridization solution containing 1 µl Ribolock RNAse inhibitor (ThermoFisher). Cells were then incubated with the probe mix for at least 16 h at 37 °C, then washed three times for 15 min at room temperature with 60% formamide, 0.3 M sodium chloride, 30 mM sodium citrate pH 7.0. Cell were then washed three times with PBS for 5 min and mounted on a slide or processed for immunofluorescence. To generate FISH probes, pCDNA3 plasmids containing viral segments were linearized with XbaI and transcribed with T7 RNA polymerase (Life Technologies) to produce a positive-sense probe to detect vRNA. Probes were directly labeled using cyanine 3- or 5-UTP (Perkin Elmer), or Chromatide Alexa Fluor 488-5-UTP (Molecular Probes)[11]. For immunofluorescence, cells were fixed for 15 min with 4% formaldehyde and permeabilized for 7 min with 0.2% (v/v) Triton-X-100 in PBS. Cells were incubated with the indicated primary antibodies for 1 h at RT, washed and incubated for 45 min with Alexa fluor conjugated secondary antibodies and Hoechst. Antibodies used were: rabbit polyclonal against Rab11a (1:100; Life Technologies, 715300), HA tag (1:500; Abcam, 9110), calnexin (1:1000, Abcam, 22595), atlastin 3 (1:100; Proteintech, 16921-1-AP), and NP (1:1000; gift from Prof Paul Digard); mouse monoclonal against NP (1:1000; Abcam, 20343), virus HA (neat; gift from Prof Paul Digard), M2 (1:500; Abcam, 5416), PDI (1:500, Life Technologies, MA3-019) and Sec31A (1:100; BD Biosciences, 612350); and goat polyclonal against ERp57 (1:200; Sicgen, AB0003-200). Secondary antibodies were all from the Alexa Fluor range (1:1000; Life Technologies). Following washing, cells were mounted with Dako Faramount Aqueous Mounting Medium and single optical sections were imaged with a Leica SP5 live confocal microscope. For cluster size quantification, images were converted to 8-bit color, background was removed, threshold adjusted and "analyze particle" function was used to determine the area of each vesicle inside selected cells. Frequency distributions were calculated and plotted with GraphPad Prism using intervals of 0–0.15, 0.15–0.30 and above 0.30 µm$^2$. Images were post-processed using Adobe Photoshop CS5 and ImageJ (NIH).

**Live-cell imaging.** Cells were grown in chambered glass-bottomed dishes (Lab-Tek) and maintained at 37 °C in Leibovitz L-15 CO$_2$-independent medium (Gibco) during imaging. Samples were imaged using Leica SP5 Inverted or Roper TIRF Spinning Disk (Yokogawa CSU-X1) and post-processed using Adobe Photoshop CS5 and ImageJ (NIH).

For FRAP analysis, cells were transfected with 250 ng of GFP-NP and immediately superinfected with PR8 at an MOI of 10. At 12 hpi, media was substituted for Leibovitz L-15 media to buffer CO$_2$ and data acquisition started on a Roper TIRF Spinning Disk (Yokogawa CSU-X1) with a cage incubator to control temperature at 37 °C. After excitation with a 491 nm laser (Cobolt 491, 100 mW), fluorescence from GFP was detected with a ×100 oil immersion objective (Plan Apo 1.49), a bandpass filter (525/45 Chroma), and a photometrics 512 EMCCD camera. All FRAP experiments were performed similarly using iLas FRAP module (Rope Scientific): 2 s prebleach, 12.18 ms µm$^{-2}$ bleach, 60 s postbleach at a frame rate of three images per second. Bleaching was performed in a variable circular area to target complete viral inclusions. For FRAP analysis, samples were corrected for background fluorescence and acquisition photobleaching as described previously by the Phair method[70]. After normalization, FRAP curves were fitted following the exponential function: $Y = Y0 + (Plateau-Y0)*(1-\exp(-D*x))$, where:
Y0: Y value when X (time) is zero. It is expressed in the same units as Y.

Plateau (must be less than one): $Y$ value at infinite times, expressed in the same units as $Y$. $D$: rate constant, expressed in reciprocal of the $x$ axis time units.

Tau: time constant, expressed in the same units as the $x$ axis. It is computed as the reciprocal of $D$.

Half time: time units of the $x$ axis. It is computed as $\ln(2)\, D^{-1}$.

Span (mobile phase): difference between $Y0$ and Plateau, expressed in the same units as your $Y$ values.

**Tokuyasu—double immunogold labeling.** Cells infected with PR8, at an MOI of 5, were fixed in suspension using 2% (v/v) formaldehyde (EMS) and 0.2% (v/v) glutaraldehyde (Polysciences) in 0.1 M Phosphate buffer (PB), for 2 h at RT. Subsequently, cells were centrifuged and washed with PB. The aldehydes were quenched using 0.15% (w/v) glycine (VWR) in 0.1 M PB for 10 min at RT. Cells were infiltrated in 12% (w/v) gelatin (Royal) for 30 min at 37 °C and centrifuged. The gelatin was solidified on ice, cut into 1 mm³ cubes and placed in 2.3 M sucrose (Alfa Aesar) in 0.1 M PB, overnight at 4 °C. The cubes were mounted onto specimen holders and frozen at −196 °C by immersion into liquid nitrogen. Samples were trimmed and cut into 50-nm-thick sections (in a Leica EM-FC7 at −110 °C) and laid onto formvar-carbon coated 100-mesh grids.

For immunogold labeling, sections were blocked with PBS/1% BSA for 20 min at RT. Antibody staining was done sequentially in PBS/1% BSA at RT: rabbit anti-GFP (1:500, 1 h), goat anti-rabbit IgG conjugated to 18 nm gold (1:20, 30 min), mouse anti-NP (1:200, 1 h), and goat anti-mouse IgG conjugated with 6 nm gold (1:20, 30 min). Gold particles were fixed by applying 1% (v/v) formaldehyde in PBS for 5 min at RT. Blocking and extensive washing were performed in-between stainings. In the final step, gold particles were fixed using 1% (v/v) glutaraldehyde (Polysciences) for 5 min RT. Grids were washed in distilled H₂O and counterstained using methyl-cellulose–uranyl acetate solution for 5 min on ice. EM images were acquired on a Hitachi H-7650 operating at 100 keV equipped with a XR41M mid mount AMT digital camera. Images were post-processed using Adobe Photoshop CS5 and ImageJ (NIH).

**Correlative light and electron microscopy (CLEM).** Cells, seeded onto gridded dishes (MatTek Corporation, P35G-2-14-C-GRID), were transfected with GFP-NP and simultaneously infected or mock-infected with PR8 at an MOI of 10. At indicated times, cells were fixed, imaged at the confocal microscope Leica SP5 Inverted and finally processed for electron microscopy imaging, as described previously[14]. Sections of 70 nm thickness were cut using a Leica EM-FC7 Ultramicrotome. The regions of interest were acquired with a Hitachi H-7650 operating at 100 keV equipped with a XR41M mid mount AMT digital camera. Images were post-processed using Adobe Photoshop CS5 and ImageJ (NIH).

**Western blotting.** Western blotting was performed according to standard procedures and imaged using a LI-COR Biosciences Odyssey near-infrared platform. Antibodies used included: rabbit polyclonal against pIRF3 (1:1000; Cell Signal, 4947), virus NP (1:1000; PB1, PB2, PA, and NS1 (all at 1:500; kindly provided by Prof. Paul Digard, Roslin Institute, UK; goat polyclonal against green fluorescent protein (GFP) (1:2000; Sicgen, AB0020), GAPDH (1:2000; Sicgen, AB0049) and virus M1 (1:500; Abcam, 20910); mouse polyclonal against virus M2 (1:500; Abcam, 5416). The secondary antibodies used were from IRDye range (1:10000; LI-COR Biosciences). The original non-cropped blots are included in the Supplementary Fig. 6 and 7.

**Enzyme-linked immunosorbent assay.** Detection of IFN-β in the cell supernatants was done using the Verikine™ Human IFN Beta ELISA kit (PBL Assay Science, 41410), range 50–4000 pg mL⁻¹, following the manufacturer's instructions.

**Quantitative real-time reverse-transcription PCR (RT-qPCR).** Extraction of RNA from samples in NZYol (NZYtech, MB18501) was achieved by using the Direct-zol RNA minipreps (Zymo Research, R2052). Reverse transcription (RT) was performed using the transcriptor first strand cDNA kit (Roche, 04896866001). Real-time RT-PCR to detect GAPDH and IFN- β, IFN-α, IL-29, and Viperin was prepared in 384-well, white, thin walled plates (Biorad, HSP3805) by using SYBR Green Supermix (Biorad, 172-5124), 10% (v/v) of cDNA and 0.4 µM of each primer. The reaction was performed on a CFX 384 Touch Real-Time PCR Detection System machine (Biorad), under the following PCR conditions: Cycle 1 (1 repeat): 95 °C for 2 min; Cycle 2 (40 repeats): 95 °C for 5 s and 60 °C for 30 s; Cycle 3: 95 °C for 5 s and melt curve 65 °C to 95 °C (increment 0.05 °C each 5 s). Data were analyzed using the CFX manager software (Biorad).

Primer sequences used for real-time RT-qPCR were the following:
GAPDH Fw: 5′-CTCTGCTCCTCCTGTTCGAC-3′;
GAPDH Rv: 5′-ACCAAATCCGTTGACTCCGAC-3′;
IL-29 Fw: 5′-AATTGGGACCTGAGGCTTCT-3′;
IL-29 Rv: 5′- GTGAAGGGGCTGGTCTAGGA-3′;
IFN-β Fw: 5′- CCTGAAGGCCAAGGAGTACA-3′;
IFN-β Rv: 5′- AAGCAATTGTCCAGTCCCAG-3′
IFN-α Fw: 5′- ATGGCCCTGTCCTTTTCTTT-3′;
IFN-α Rv: 5′- ATTCTTCCCATTTGTGCCAG-3′

Viperin Fw: 5′- TCACTCGCCAGTGCAACTAC-3′
Viperin Rv: 5′- TGGCTCTCCACCTGAAAAGT-3′

**Reporting summary.** Further information on experimental design is available in the Nature Research Reporting Summary linked to this article.

## Data availability

The authors declare that the data supporting the findings of this study are available within the paper and its Supplementary Information files.

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

## Acknowledgements

This project is supported by the grant PTDC/BIA-CEL/32211/2017 awarded by the Fundação para a Ciência e a Tecnologia (FCT, Portugal) and also by Fundação Calouste Gulbenkian-Instituto Gulbenkian de Ciência (FCG-IGC, Portugal). A.L.S. and F.F. are funded by FCG. All other authors are supported by FCT: S.V.-C. and M.A. are funded by post-doctoral working contracts; T.A.E. is supported by the PhD fellowship PD/BD/128436/2017 and M.J.A. is funded by the FCT investigator contract IF/00899/2013. The authors acknowledge Prof Paul Digard (Roslin Institute, UK), Dr. Ron Fouchier (Erasmus, Netherlands), Dr. Colin Adrain (IGC, Portugal), and Dr. Christoph Dehio (University of Basel, Switzerland) for providing cells and reagents. We are grateful to the Unit of Imaging and Cytometry at the Instituto Gulbenkian de Ciência for technical support, sample processing and data collection. The authors thank Dr. Fabrice Cordelières from the Bordeaux Imaging Center (INSERM, France), Dr. Erin Tranfield, Dr. Gabriel Martins, Nuno Pimpão, and Dr. Luís Moita (all from IGC, Portugal) for helpful discussion.

## Author contributions

M.J.A., M.A., and S.V.-C.: designed the experiments; M.J.A., M.A., S.V-C., T.A.E., F.F., and A.L.S.: carried out experiments and analyzed the data; M.J.A. conceived and supervised the research; M.J.A., M.A., and S.V.-C.: wrote the manuscript; and all authors contributed to editing the manuscript.

## Additional information

**Competing interests:** The authors declare no competing interests.

