## [Peer Review File · Nature Communications]

Reviewers' comments:

Reviewer #1 (Remarks to the Author):

Influenza A virus ribonucleoproteins form liquid organelles at endoplasmic reticulum exit sites

Alenquer et al.,

In this study the authors shed new light on how Rab11 and ER mediate influenza A virus (IAV) vRNP egress. The authors show that assembling vRNPs form viral inclusions that have liquid organelle-like properties that resemble stress granules, PML bodies etc. These inclusions can be disassembled by hypotonic shock. Viral inclusion formation is dependent on Rab11 activity and Golgi-ER traffic which can be blocked by Brefeldin A. They show that viral inclusions form in the ER proximity but do not overlap with the ER - it is unclear whether the vRNPs motility is generated. ER exit sites (ERES) colocalize with vRNPs and serve as docking sites or portals where vRNPs primarily accumulate. RNA-RNA interactions are not essential for the viral inclusions to form, because a single segment vRNP is sufficient to induce inclusions. Overexpression of Sar1-GTP and blockage of Golgi-ER traffic reduced the size of viral inclusions and number. Accordingly the size and number of inclusions was reduced by BFA treatment. BFA disassembled the viral inclusions in less than 1 hour post addition indicating that inclusions are dynamic. NS1 delta PR8 was compared with WT PR8 in Rab11 WT or DN expressing cells, to conclude that innate immune responses are unaffected by phase transitions of vRNPs. The study is well-written and presented. The movies and image data are appropriately quantified. Although mechanistic insights are lacking the phenotype is well documented.

Major points:

1) Fig1 and FigS1 How do the authors know that all GFP-NP puncta are viral inclusions? Are the quantifications derived from CLEM or just IFA? Perhaps a more precise definition of 'viral inclusion' is necessary whether it is morphological (EM), FISH-positive, or a fluorescence NP signal above a certain threshold. The authors must rule out that these are monomeric GFP-NP. As a negative control, they could also test a mutant NP sequence (e.g. from bat IAV NP) which is not incorporated into vRNPs (Moreira et al., 2016, Nat Comm).

2) Do the authors see structures resembling the ICVs (irregularly coated vesicles) (Castro-Martin et al., 2017, Nat Comm)?

3) It is important to demonstrate that the GFP-NP viral inclusions are intermediates that lead to a productive infection. Are the GFP-NP packaged into progeny virions i.e. do they traffic to viral budding sites by CLEM/IFA, are they detected in released viral particles (e.g. ultracentrifugal sedimentation), and co-label by FISH?

4) The WSN PB2-GFP strain (Lakdawala et al., 2014, PLoS Path) has been used with live light-sheet microscopy to show that vRNP intermediates fuse in the cytoplasm during egress. Since the GFP is fused to PB2 (and not overexpressed) the strain would provide a more direct way to confirm the liquid inclusion phenomenon.

4) Can the authors use viable packaging-deficient IAV mutants (e.g. Hutchinson et al., 2009); to test the RNA-RNA complementary interactions essential for the packaging of the eight individual segments?

5) Fig.1c Measurements of the nuclear vRNP signal intensity should be shown as a comparison. Do nuclear GFP-NP spots have similar properties to cytoplasmic viral inclusions? Should they not only form inclusions after nuclear export? FISH should be done to show the identity of the GFP-NP spots.

5) Fig.2 Since microtubule motors are known to regulate ER-Golgi transport, it would be interesting to know how the events shown in Fig. 2c, d, e, f are impacted by nocodazole. Similarly, BFA, kinesin or dynein inhibitors could be tested.

6) Fig.4c The reduction in viral titer (WT vs NS1) is greater in the Rab11DN cells than the Rab11WT. At 16h the reduction is ca 50% (Rab11WT) vs 75% (Rab11DN). When viral inclusions are not able to form, is IAV more sensitive to the antiviral immune response?

7) Fig. 6 The authors provide a connection between viral inclusions and ER exit sites. What is the impact of Sar1-GTP expression (or Sar1 depletion) on viral titre? The authors could perform dual/triple fluorescent CLEM to examine the ultrastructural signatures of ERES/inclusions in more depth.

Minor points:

1) Fig.1b Arrow colours: should be made distinguishable for colour-blind people.

2) Fig.1. It would be good to have a control experiments using GFP or a GFP-NP mutant that does not form GFP-positive viral inclusions.

3) Fig.1c Do other destabilisers of liquid inclusions e.g. heat, chemotoxicity, have the similar end effect as hypotonic shock?

4) Fig.6c Are these images z-stack projections?

5) line 283: It is noteworthy that....(or similar)

Further comments:

It is well known that the live tracking of infectious vRNPs is a difficult task due to the sensitive nature of flu viral core components to genomic editing.

Because of this, the characterisation and verification of GFP-NP punctae as infectious vRNP constituents is absolutely crucial.

In agreement with Referee #2's comments, the authors should provide more data to show that GFP-NP is truly a part of infectious vRNPs (such as by FISH or biochemistry of purified virions). A considerable number of GFP molecules are needed in order to visualise GFP by time lapse microscopy and therefore raises concerns about artefacts of the overexpression approach. They could additionally use the WSN PB2-GFP strain (Lakdawala et al., 2014) to complement their approach, as I have additionally suggested.

The use of a (viable) packaging signal defective mutant is also useful, to test whether GFP-NP motility and fusion is dependent on RNA-RNA interactions between the individual vRNP segments as expected, and/or as negative control to use GFP-NP mutants that are not packaged into vRNPs.

I feel there are several straightforward approaches that the authors have neglected, but will considerably increase the validity of their claims when performed appropriately. In other words controls are lacking. A recent paper (Castro-Martin et al., 2017) looks at a similar aspect of flu vRNP assembly. Furthermore, mechanistic insight is lacking in both studies (though one is already

published). Thus, the liquid inclusion hypothesis is the only new aspect, if I may so, that makes the study interesting. Thus, I think Dr. Amorim and co. must increase the quality of the validation of their hypothesis for it to be publishable (as a considerable advancement to the influenza field) in Nature Communications.

Reviewer #2 (Remarks to the Author):

The manuscript „Influenza A virus ribonucleoproteins form liquid organelles at endoplasmic reticulum exit sites” by Alenquer et al. investigates viral inclusions formed upon influenza A virus infection.

The paper is well written and addresses complex questions regarding the cellular rearrangements preceding the assembly of progeny influenza A virus particles. However, several questions are still unclear and need further investigations:

- 1) Several sentences including a large proportion of the abstract, in particular the assembly process of vRNPs en route to the plasma membrane are presented as given fact rather than hypotheses and should be phrased more carefully. Has it ever been mentioned that RNA-RNA interactions are the “driving force” of punctate structures upon infection?
- 2) It remains unclear, whether the GFP-NP construct is part of functional vRNPs exported to the cytoplasm. This issue should be addressed. Specifically, are GFP-NP molecules functional and support the polymerase function or do they abrogate vRNP formation. The latter might compromise the conclusions.
- 3) The authors typically claim that their inclusion bodies contain vRNPs. However, except for Fig.3c, they exclusively visualize transfected NP rather than vRNPs derived from infection.
- 4) A viral growth curve (MOI: 0.001) in Rab11 KO cells should be included.
- 5) A double infection using two different virus isolates followed by a simultaneous FISH staining of the same segment could help to clarify whether these inclusion bodies represent assembly sites for vRNPs. This would have a significant effect on the discussion.
- 6) Fig. 1a: Do the authors observe similar inclusion bodies in WT infected cells without GFP-NP transfection?

7) Fig. 1a: It remains unclear, whether these cells are really infected. A double staining using either FISH probes specific for certain segments or a PB2 protein staining would help to distinguish transfected, non-infected cells from cells positive for both transfection and infection.

8) Fig. 1a: a Mock-infected cell, likewise transfected with GFP-NP is missing.

9) Fig. 1a: which timepoint post infection?

10) Fig. 2a: Again, it remains unclear whether this cell is infected or not. A double staining is necessary. Likewise a negative control is missing.

11) Fig. 2b: How do the authors explain the differences in the photobleached regions?

12) Fig. 3: The title is misleading. How do the authors exclude that there are RNA-RNA interactions among neighbouring vRNPs comprising the same vRNA? This has not been shown.

13) Fig. 4b+c: The biological significance is unclear. At 8 hpi, both PR8 WT infected WT and DN cells have comparable viral titers (4C), yet there is a striking difference in the size of the inclusion bodies (4B).

14) Fig 4c: the scale is misleading. Please use a log scale showing viral titers.

15) Fig. 6b: Could the authors provide an analysis for Mock-infected cells. Does the size of inclusion bodies similarly differ?

ANSWERS TO REVIEWERS' COMMENTS:

Please note that given the extent of the alterations, we decided to include two manuscripts, one of them without highlighting the changes (Manuscript) and another that includes all alterations (Supplementary file 2).

Reviewer #1 (Remarks to the Author):

Influenza A virus ribonucleoproteins form liquid organelles at endoplasmic reticulum exit sites, Alenquer et al.,

In this study the authors shed new light on how Rab11 and ER mediate influenza A virus (IAV) vRNP egress. The authors show that assembling vRNPs form viral inclusions that have liquid organelle-like properties that resemble stress granules, PML bodies etc. These inclusions can be disassembled by hypotonic shock. Viral inclusion formation is dependent on Rab11 activity and Golgi-ER traffic which can be blocked by Brefeldin A. They show that viral inclusions form in the ER proximity but do not overlap with the ER - it is unclear whether the vRNPs motility is generated. ER exit sites (ERES) colocalize with vRNPs and serve as docking sites or portals where vRNPs primarily accumulate. RNA-RNA interactions are not essential for the viral inclusions to form, because a single segment vRNP is sufficient to induce inclusions. Overexpression of Sar1-GTP and blockage of Golgi-ER traffic reduced the size of viral inclusions and number. Accordingly the size and number of inclusions was reduced by BFA treatment. BFA disassembled the viral inclusions in less than 1 hour post addition indicating that inclusions are dynamic. NS1 delta PR8 was compared with WT PR8 in Rab11 WT or DN expressing cells, to conclude that innate immune responses are unaffected by phase transitions of vRNPs. The study is well-written and presented. The movies and image data are appropriately quantified. Although mechanistic insights are lacking the phenotype is well documented.

Major points:

1) Fig1 and FigS1 How do the authors know that all GFP-NP puncta are viral inclusions? Are the quantifications derived from CLEM or just IFA? Perhaps a more precise definition of 'viral inclusion' is necessary whether it is morphological (EM) or a fluorescence NP signal above a certain threshold.

---- We have defined more precisely viral inclusions in three different infection systems: infection; transfection of GFP-NP and simultaneous infection; and infection with a productive virus encoding a PA-GFP. By fluorescence in situ hybridization of several segments and immunofluorescence, we show that NP, GFP-NP (when applicable), PA (when applicable) co-localize with several RNA segments in the cytosol. These are fully formed RNPs as globally are always found associated to each other. This data has now been included in Figure 2.

2) Do the authors see ICVs (irregularly coated vesicles) postulated by Castro-Martin et al., (2017)? Please discuss.

---- Yes, we do see the ICVs by several techniques including electron microscopy, electron tomography, CLEM and immunogold labelling. In this manuscript, we showed the immunogold labelling data in Figure 1a and 1b that has now been renumbered to Figure 4. In the JCS paper we published in 2016 with the doi: 10.1242/jcs.188409 (<https://doi.org/10.1242/jcs.188409>), we have shown the ICVs by CLEM (Figures 5-7, and supplementary Figure S3-S4). We called these structures heterogeneous clustered vesicles and, in 2017, the de Castro Martin Nature

Communication paper renamed these structures to ICVs. For the sake of clarity, we included a sentence to mention that clustered vesicles and ICVs are the same thing.

3) It is important to demonstrate that the GFP-NP viral inclusions are intermediates that lead to a productive infection. Can the authors show that GFP-NP is packaged into progeny virions i.e. they locate at viral budding sites by CLEM or IFA, and are detected in released viral particles?

--- We agree with reviewers that it is important to show that tagged RNPs result in productive infections.

In two previous manuscripts, using a minireplicon system to produce GFP-tagged RNPs (doi:10.1128/JVI.02606-10) and in viral co-infection in the presence of GFP-NP (doi:10.1128/JVI.03123-12), we showed that productive RNPs carrying GFP-NP are formed - (See below, at the end of this point the full explanation of the findings).

We have now decided that our best option would be to show viral inclusions with liquid properties in another system and have opted for cells infected with a productive virus encoding PA-GFP. By live cell imaging, we show that the properties of viral inclusions in the GFP-NP co-infection system and the PA-GFP encoding virus are similar: both contain concentrated RNPs, dissolve upon hypotonic shock and hexanediol, and undergo fusion and fission events. In addition, we have combined the two techniques: the PA-GFP virus with a cherry-NP and have observed cytosolic structures with the same characteristics that, in addition, accumulate GFP and Cherry. All the data has been added to Figures 2 and 3.

Furthermore, we have extensively characterized viral inclusions, showing that GFP-NP in the cytosol is incorporated in RNPs, as it colocalizes with the RNA of several segments, untagged NP and PA, indicating that these are complete RNPs.

Finding GFP-tagged RNPs in the cytosol (in structures that contain viral RNA, NP and PA, both in the GFP-NP co-transfection system and upon PA-GFP viral infection) also means that GFP-NP or cherry-NP transfection and co-infection supports the polymerase function. However, to comply with this reviewers' request, we showed that GFP-NP or cherry-NP does not inhibit viral polymerase as evaluated using a mini genome reporter plasmid that produces a negative-sense luciferase gene bounded by the viral promoter sequences. Results have been added to Figure 1.

We have therefore strong evidence to say that GFP-NP, when in the cytosol, is incorporated in RNPs, and we find the same properties in viral inclusions formed by a productive virus encoding PA-GFP that permits live cell imaging. We agree that including these additional data makes the conclusions from the manuscript stronger.

COMPLETE EXPLANATION OF PREVIOUSLY PUBLISHED DATA:

In the manuscript of 2011, published in Journal of Virology with doi:10.1128/JVI.02606-10 (<https://doi.org/10.1128/JVI.02606-10>), Amorim MJ (the senior author of the present work) et al., used a mini-replicon-system to produce and track vRNPs tagged with GFP by using a mixture of 4:1 of plasmids encoding NP:GFP-NP. In Figure 5 of that manuscript, Amorim et al., characterized that RNPs incorporate GFP-NP by immunoprecipitation and by FISH, in which GFP-NP in the cytosol colocalizes with the RNA of several segments. This finding validated the method, as NP can only be found in the cytosol if in a complex with RNPs. NP contains three nuclear localization signals, and hence relocates to the nucleus upon translation.

Amorim then proceeded to using GFP-NP in infected cells to mark and track RNPs. In 2013, a manuscript was published in the Journal of Virology with doi:10.1128/JVI.03123-12 (<https://doi.org/10.1128/JVI.03123-12>) showing biochemically that GFP-NP is incorporated in RNPs in infected cells. Figure 3 shows an immunoprecipitation of GFP-NP, together with PB1, PB2, RNA of segment 3, NP (and NP-GFP). However, no evidence was provided that these RNPs reached the cytosol, which has now been added to Figure 2 of this manuscript by doing FISH in the 3 systems: GFP-NP transfection and co-infection, PA-GFP encoding virus and infection.

4) Fig.1c Measurements of the nuclear vRNP signal intensity should be shown as a comparison. Do nuclear vRNP spots have similar properties to viral inclusions in the cytoplasm? Please discuss.
--- We have done what was requested and here are the results and our interpretation:

Figure 1 - A549 cells transfected with GFP-NP and infected or mock infected and imaged under time lapse conditions upon adding media or 80% water (hypotonic shock) for 10 min. Images were fixed, and integrated density of NP determined in the nucleus and cytosol and values of 12 cells were plotted. Statistical analysis of data was performed using two-way ANOVA, followed by Tukey's multiple comparisons test (* $p < 0.01$; ** $p < 0.05$).

GFP-NP accumulates in large amounts in the cell nucleus, because it contains nuclear localization signals (Nakada et al., 2015) and is only detected in the cytosol when in a complex with RNPs. In the nucleus, GFP-NP accumulates in specific regions and therefore we asked if the nuclear NP accumulations were sensitive to hypotonic shock. We calculated integrated density of GFP-NP in mock and infected cells before and after hypotonic stress. Mock cells have a higher accumulation of GFP-NP in the nucleus and upon infection a significant reduction of GFP-NP in the nucleus is accompanied by an increase of this protein in the cytosol. However, upon hypotonic stress, whilst the cytosolic GFP-NP integrated density significantly decreased, on account of dissolving viral inclusions, the value of nuclear GFP-NP did not change, regardless of being infected or mock infected, showing that nuclear GFP-NP is resistant to the 10 minutes of hypotonic stress applied. This might reflect the time the dissolved media takes to access the nucleus, but it could similarly translate differences in the type of nuclear viral inclusions as the interactions of vRNPs with host proteins in the nucleus and cytosol are different. Further experiments should clarify this point
Upon careful assessment, we decided not to include the result in the manuscript because we reasoned that it interrupts the line of thought. If this reviewer thinks strongly otherwise, we are of course willing to include it.

5) Fig.2 Since microtubule motors are known to regulate ER-Golgi transport, it would be interesting to know how the events shown in Fig. 2c, d, e, f are impacted by nocodazole. Similarly, BFA, kinesin or dynein inhibitors could be tested.

---- We have tested BFA in live cells and have included it in Figure 6. In this movie, the viral inclusions dissolve very quickly, but during the time we can image them, they fuse and suffer fission events. As requested we calculated viral titers in the presence and absence of BFA and have included this in Figure 6. Interestingly, BFA is the drug that, in our hands, reduces the most viral infection (compared with microtubule or actin-affecting drugs and monesin (see ref Doi: JVI.02606-

10 [pii] 10.1128/JVI.02606-10). However, the drug impacts in multiple viral steps including HA, NA and M2 transport and at the moment it is not possible to accurately evaluate its effect on RNP complex assembly individually.

In addition, we evaluated fusion and fission events in the presence of nocodazole and latrunculin A in the GFP-NP co-infection system and in the PA-GFP encoding virus. We included this data in Figure 2. Nocodazole did not block viral inclusion formation, but it severely impaired fusion and fission events. Conversely latrunculin A had no effect.

6) Fig.4c The reduction in viral titer (WT vs NS1) is greater in the Rab11DN cells than the Rab11WT. At 16h the reduction is ca 50% (Rab11WT) vs 75% (Rab11DN). When viral inclusions are not able to form, is IAV more sensitive to the antiviral immune response?

---- Our results indicate that IFN response is similar in Rab11WT and in Rab11DN cells upon viral infection, and thus far we have no reason to suspect that IAV is more sensitive to antiviral immune response in the absence of Rab11.

7) Fig. 6 The authors provide a connection between viral inclusions and ER exit sites. What is the impact of Sar1-GTP expression (or Sar1 depletion) on viral titre? The authors could perform dual/triple fluorescent CLEM to examine the ultrastructural signatures of ERES/inclusions in a little more depth.

---- As mentioned, in our JCS manuscript of 2016 with the doi: 10.1242/jcs.188409 we have identified that Rab11 vesicles cluster upon infection with influenza A virus. These clusters were denominated heterogeneous clustered vesicles as they varied in size, but also in nature, presenting vesicles with single and double membranes. We now know that double membranes are derived from the ER and we are carrying a project in the lab looking at alterations of the ER during infection (unrelated to those described in the nature communications paper with doi 10.1038/s41467-017-01557-6). We are depleting several ER factors and analyzing the impact in viral inclusions and viral titers. These studies are accompanied with CLEM and electron tomography and aim to analyze the ultrastructural signatures of viral inclusions. The results we are acquiring show complex architectural changes that we are investigating in an independent study. Therefore, at present we prefer not to disclose these findings. I hope you can understand our reasoning.

Minor points:

1) Fig.1b Arrow colours: should be made distinguishable for colour-blind people.

---- Thank you for this remark and it has been done.

2) Fig.1. It would be good to have a control experiments using GFP or a GFP-NP mutant that does not form GFP-positive viral inclusions.

In the JCS paper we published in 2016 with the doi: 10.1242/jcs.188409 (<https://doi.org/10.1242/jcs.188409>), we have shown viral inclusions in cells that do not express GFP-NP, and we think it is not necessary to duplicate similar results.

3) Fig.1c Do other destabilisers of liquid inclusions e.g. heat, chemotoxicity, have the similar end effect as hypotonic shock?

---- We have tested the effect of 1, 6 - hexanediol in GFP-NP co transfection system and in PA-GFP expressing viruses and show that the effect of hexanediol dissolves viral inclusions, although the effect is less pronounced than the hypotonic shock (Figure 3).

4) Fig.6c Are these images z-stack projections?

--- No, images are single XY planes, but the differences are very striking looking over the microscope and are recapitulated in this panel.

5) line 283: It is noteworthy that....(or similar).
---- We thank this reviewer for pointing this out.

Reviewer #2 (Remarks to the Author):

The manuscript „Influenza A virus ribonucleoproteins form liquid organelles at endoplasmic reticulum exit sites” by Alenquer et al. investigates viral inclusions formed upon influenza A virus infection.

The paper is well written and addresses complex questions regarding the cellular rearrangements preceding the assembly of progeny influenza A virus particles. However, several questions are still unclear and need further investigations:

1) Several sentences including a large proportion of the abstract, in particular the assembly process of vRNPs en route to the plasma membrane are presented as given fact rather than hypotheses and should be phrased more carefully. Has it ever been mentioned that RNA-RNA interactions are the “driving force” of punctate structures upon infection?

--- The reviewer is correct and we have decided to rephrase the abstract as requested.

2) It remains unclear, whether the GFP-NP construct is part of functional vRNPs exported to the cytoplasm. This issue should be addressed. Specifically, are GFP-NP molecules functional and support the polymerase function or do they abrogate vRNP formation. The latter might compromise the conclusions.

--- We agree with reviewers that it is important to show that GFP-NP originates functional RNPs and support the polymerase function. This answer is the same as provided for reviewer #1.

In two previous manuscripts, using a minireplicon system to produce GFP-tagged RNPs (doi:10.1128/JVI.02606-10) and in viral co-infection in the presence of GFP-NP (doi:10.1128/JVI.03123-12), we showed that productive RNPs carrying GFP-NP are formed (by immunoprecipitation) - (See at the end of point 3) the full explanation of the findings). However, we did not attempt to show that GFP-NP was incorporated in virions.

To comply with reviewers questions, we have opted by showing that viral inclusions with liquid properties are formed in another system: in cells infected by a productive virus encoding PA-GFP. By live cell imaging, we showed that the properties of viral inclusions in the GFP-NP co-infection system and the PA-GFP encoding virus are similar: both dissolve upon hypotonic shock and hexanediol, fuse and suffer fission events. In addition, we have combined the two techniques: the PA-GFP virus with a cherry-NP and this strategy also gives rise to cytosolic structures with the same characteristics that in addition accumulate GFP and cherry in the same structures. All the data has been added to Figure 2 and 3.

Furthermore, we have extensively characterized viral inclusions, showing that GFP-NP in the cytosol is incorporated with RNPs as it colocalizes with the RNA of several segments, untagged NP and PA, indicating that these are complete RNPs.

Finding GFP-tagged RNPs in the cytosol (in structures that contain viral RNA, NP and PA both in the GFP-NP co-transfection system and upon PA-GFP viral infection) also means that GFP-NP or cherry-NP transfection and co-infection supports the polymerase function. However, to comply with this reviewers' request, we showed that GFP-NP or cherry-NP does not inhibit viral polymerase as evaluated using a mini genome reporter plasmid that produces a negative-sense luciferase gene bounded by the viral promoter sequences. Results have been added to the manuscript as supplementary Figure 1.

We have therefore strong evidence to say that GFP-NP, when in the cytosol, is incorporated in RNPs, but above all we find the same properties in viral inclusions formed by a productive virus

encoding PA-GFP that permits live cell imaging. We agree that including these additional data makes the conclusions from the manuscript stronger.

COMPLETE EXPLANATION OF PREVIOUSLY PUBLISHED DATA:

In the manuscript of 2011, published in Journal of Virology with doi:10.1128/JVI.02606-10 (<https://doi.org/10.1128/JVI.02606-10>), Amorim MJ (the senior author of the present work) et al., used a mini-replicon-system to produce and track vRNPs tagged with GFP by using a mixture of 4:1 of plasmids encoding NP:GFP-NP. In Figure 5 of that manuscript, Amorim et al., characterized that RNPs incorporate GFP-NP by immunoprecipitation and by FISH, in which GFP-NP in the cytosol colocalizes with the RNA of several segments. This finding validated the method as NP can only be found in the cytosol if in a complex with RNPs. NP contains three nuclear localization signals, and hence relocates to the nucleus upon translation.

Amorim then proceeded to using GFP-NP in infected cells to mark and tracks RNPs. In 2013, a manuscript was published in the Journal of Virology with doi:10.1128/JVI.03123-12 (<https://doi.org/10.1128/JVI.03123-12>) showing biochemically that GFP-NP is incorporated in RNPs in infected cells. Figure 3 shows an immunoprecipitation of GFP-NP, together with PB1, PB2, RNA of segment 3, NP (and NP-GFP). However, no evidence was provided that these RNPs reached the cytosol, which has now been added to Figure 1 of this manuscript by doing FISH in the 3 systems: GFP-NP transfection and co-infection, PA-GFP encoding virus and infection.

3) The authors typically claim that these inclusion bodies contain vRNPs. However, except for Fig.3c, they exclusively visualize transfected NP rather than vRNPs derived from infection.

--- We have added Figure 2 to show GFP-NP colocalizes with several viral RNAs and the viral polymerase PA. By doing so, in the subsequent figures we can use GFP-NP as a proxy for RNPs.

4) A viral growth curve (MOI: 0.001) in Rab11 KO cells should be included.

--- We used cell lines expressing Rab11-GFP wild-type and dominant negative forms that were sorted for high and low expression of Rab11. The data shows that cell lines expressing low levels of Rab11 WT produce similar viral titres as the control, but those in the same conditions for Rab11 DN have 2 logs decrease in viral production (Supplementary Fig 5). As the levels of expression rise for Rab11WT, and the viral inclusion size increases, viral production is also affected. Our data indicates that the size of viral inclusions needs to be within a specific range to favor infection. Above or below that optimal range we observe a detrimental effect in viral replication. We are currently establishing the conditions of viral inclusions that favor infection in an independent project.

5) A double infection using two different virus isolates followed by a simultaneous FISH staining of the same segment could help to clarify whether these inclusion bodies represent assembly sites for vRNPs. This would have a significant effect on the discussion.

--- We have done the suggested experiment and observed that segment 4 and 6 of two parental viruses, which are significantly different and easier to distinguish by FISH, colocalize in viral inclusions (Figure 9). We have elaborated on the findings in the discussion.

6) Fig. 1a: Do the authors observe similar inclusion bodies in WT infected cells without GFP-NP transfection?

--- In the JCS paper we published in 2016 with the doi: 10.1242/jcs.188409 (<https://doi.org/10.1242/jcs.188409>), we have shown viral inclusions in cells that do not express GFP-NP, and we think it is not necessary to duplicate similar results.

7) Fig. 1a: It remains unclear, whether these cells are really infected. A double staining using either FISH probes specific for certain segments or a PB2 protein staining would help to distinguish transfected, non-infected cells from cells positive for both transfection and infection.

--- In the electron microscopy images we specifically used cells that had virions budding. These are well characterized and specific events to influenza infected cells. Budding virions, have an established height of 80-120 nm, around 40-60 nm width and contain RNPs inside and are impossible to confuse by other structure.

8) Fig. 1a: a Mock-infected cell, likewise transfected with GFP-NP is missing.

--- We have added the mock infected cell to the supplementary Figure 3.

9) Fig. 1a: which timepoint post infection?

--- The timepoint is 16 h p.i. and we added the information to the figure legend.

10) Fig. 2a: Again, it remains unclear whether this cell is infected or not. A double staining is necessary. Likewise a negative control is missing.

--- We included negative controls in Fig 2c, lower right panel, Fig 5D and in supplementary Fig 3. Infected and not infected cells transfected with GFP-NP are readily distinguishable in terms of NP localization. In uninfected cells, NP is restricted to the nucleus, as NP contains three nuclear localization signals. In infected cells, NP is found both in the nucleus and the cytosol. NP found in the cytosol is part of a RNP (that hides NLSs).

11) Fig. 2b: How do the authors explain the differences in the photobleached regions?

--- The exchange of material of each viral inclusion is unique. As we observed and have included in the text (Fig. 4b-c) some inclusions receive incoming material, but others just donate their own material, which gives rise to a FRAP profile that has a very high standard deviation. Other alternatives include that few viral inclusions in cells transit to a more gel like phase. This result is also consistent with the hexanediol data. However, if it happens, it would be in a minority of inclusions.

12) Fig. 3: The title is misleading. How do the authors exclude that there are RNA-RNA interactions among neighboring vRNPs comprising the same vRNA? This has not been shown.

--- It has been shown that defective interfering particles, that lack a big internal portion of the segments, but contain both termini, outcompete the same segment for packaging in virions (Davis et al., 1980, Duhaut and Dimmock, 2000, Duhaut and Dimmock, 1998, Duhaut and Dimmock, 2002, Duhaut and McCauley, 1996). For an influenza particle to be fully infectious the eight RNPs must be packaged in a virion. Virions do not usually package more than eight segments and each segment usually only occurs once per virion. Given the amount of data corroborating these observations, one can safely assume that RNPs from the same segment do not interact in the context of infection.

13) Fig. 4b+c: The biological significance is unclear. At 8 hpi, both PR8 WT infected WT and DN cells have comparable viral titers (4C), yet there is a striking difference in the size of the inclusion bodies (4B).

--- As stated in point 4, our data indicates that the size of viral inclusions needs to be within a specific range to favor infection. Above or below that optimal range we observe a detrimental effect in viral replication. In accordance with this, we have included viral titrations in cells expressing low or high levels of Rab11DN and Rab11WT. For the question of innate immunity activation, in the form of IFN expression and release, all we cared about was the size of viral inclusions and not the effect on viral titres. Therefore, to maximize the size of viral inclusions we used cell lines mixed for high and

low expression of the protein in question. This has been explained when describing Supplementary Figure 5).

14) Fig 4c: the scale is misleading. Please use a log scale showing viral titers.

--- We have done so. In all our experiments similar ratios between DN and WT samples were always obtained, however variability between experiences was high, and that is the reason for using percentages. Therefore we repeated the experiment, increasing the N and analysed viral titres in triplicate. The result we present is representative of 3 experiments with similar ratios/differences between viruses grown in WT or DN cells. We opted for this solution to overcome high variability between experiments. Small differences in cell seeding, time of infection, time of incubation can be translated into high experimental variance.

15) Fig. 6b: Could the authors provide an analysis for Mock-infected cells. Does the size of similarly differ?

--- Inclusion bodies do not form in mock-infected cells and are only a component in infected cells. The number of inclusions was determined by measuring NP staining, that is only expressed in infected cells. We have however looked at the cellular component found in viral inclusions in infection, which is Rab11, and we see that this does not change if cells are not infected (see below). Of note, changes in Rab11 are not unexpected and corroborate previously reported findings that Rab11 pathway is modified upon IAV challenged.

Figure 2 - A549 cells were transfected with GFP or SAR1-GTP and infected or mock infected with PR8 at an MOI of 5. 16 hpi, cells were fixed and stained for NP and Rab11. Note that Rab11 distribution changes with infection but not with SAR1-GTP. The experiment was done thrice.

REFERENCES:

DAVIS, A. R., HITI, A. L. & NAYAK, D. P. 1980. Influenza defective interfering viral RNA is formed by internal deletion of genomic RNA. *Proc Natl Acad Sci U S A*, 77, 215-9.

- DUHAUT, S. & DIMMOCK, N. J. 2000. Approximately 150 nucleotides from the 5' end of an influenza A segment 1 defective virion RNA are needed for genome stability during passage of defective virus in infected cells. *Virology*, 275, 278-85.
- DUHAUT, S. D. & DIMMOCK, N. J. 1998. Heterologous protection of mice from a lethal human H1N1 influenza A virus infection by H3N8 equine defective interfering virus: comparison of defective RNA sequences isolated from the DI inoculum and mouse lung. *Virology*, 248, 241-53.
- DUHAUT, S. D. & DIMMOCK, N. J. 2002. Defective segment 1 RNAs that interfere with production of infectious influenza A virus require at least 150 nucleotides of 5' sequence: evidence from a plasmid-driven system. *J Gen Virol*, 83, 403-11.
- DUHAUT, S. D. & MCCAULEY, J. W. 1996. Defective RNAs inhibit the assembly of influenza virus genome segments in a segment-specific manner. *Virology*, 216, 326-37.
- NAKADA, R., HIRANO, H. & MATSUURA, Y. 2015. Structure of importin- α bound to a non-classical nuclear localization signal of the influenza A virus nucleoprotein. *Scientific Reports*, 5, 15055.

REVIEWERS' COMMENTS:

Reviewer #1 (Remarks to the Author):

Marta Alenquer, Sílvia Vale-Costa et al.

INFLUENZA A VIRUS RIBONUCLEOPROTEINS FORM LIQUID ORGANELLES AT 2 ENDOPLASMIC RETICULUM EXIT SITES

NCOMMS-18-24184B

The authors have addressed all the reviewers' concerns point by point. The authors showed GFP-NP incorporation into vRNPs, somewhat a concern by both reviewers, in a previous mini-replicon experiment.

The study is thoroughly performed with appropriate quantifications and controls. As a result I believe there is significant cell biological insight into influenza vRNP budding. I believe this manuscript will be ready for publication after making some small editorial corrections below.

Minor corrections:

-Line 137; cherry-NP > Cherry-NP

Unify the name with in lines 139, 140 and Fig.1 legend.

-Line 155; (Fig. 3e, lower panels), > (Fig. 1e, lower panels),

-Line 203; cherry-NP > Cherry-NP

-Line 204; cherry-NP > Cherry-NP

-Fig.2b; mCherry-NP > Cherry-NP

-Line 228 (Fig.2b legend); GFP-NP/GFP > Cherry-NP? Please make sure this is correct.

-Line 288 (Fig.3c legend); c. > remove

-Line 362; (white arrowheads and small dots) > (white arrowheads and small dots, Fig. 4e)

-Fig. 5c; Sec31A > Sec31

-Line 424; GM310 > GM130

-Line 516 & 517; Fig. 8d > Fig. 8e, Fig.8e > Fig. 8d

-Line 535; c. > c. (bold)

-Line 539 & 540 (Fig. 8 legend); swap d for e and e for d.

Reviewer #2 (Remarks to the Author):

The paper significantly improved and the reviewer addressed all open issues.

RESPONSE TO REFEREES:

This document file contains our point-by-point response to REVIEWERS' COMMENTS.

NCOMMS-18-24184B

Marta Alenquer, Sílvia Vale-Costa et al. INFLUENZA A VIRUS RIBONUCLEOPROTEINS FORM LIQUID ORGANELLES AT ENDOPLASMIC RETICULUM EXIT SITES

Reviewer #1 (Remarks to the Author):

The authors have addressed all the reviewers concerns point by point. The authors showed GFP-NP incorporation into vRNPs, somewhat a concern by both reviewers, in a previous mini-replicon experiment. The study is thoroughly performed with appropriate quantifications and controls. As a result I believe there is significant cell biological insight into influenza vRNP budding. I believe this manuscript will be ready for publication after making some small editorial corrections below.

Minor corrections:

-Line 137; cherry-NP > Cherry-NP

Unify the name with in lines 139, 140 and Fig.1 legend.

We have unified the names cherry-NP and Cherry-NP to mCherry-NP throughout the entire manuscript and supplementary information

-Line 155; (Fig. 3e, lower panels), > (Fig. 1e, lower panels),

We have corrected this error

-Line 203; cherry-NP > Cherry-NP

We have unified the names cherry-NP and Cherry-NP to mCherry-NP throughout the entire manuscript and supplementary information

-Line 204; cherry-NP > Cherry-NP

We have unified the names cherry-NP and Cherry-NP to mCherry-NP throughout the entire manuscript and supplementary information

-Fig.2b; mCherry-NP > Cherry-NP

We have unified the names cherry-NP and Cherry-NP to mCherry-NP throughout the entire manuscript and supplementary information

-Line 228 (Fig.2b legend); GFP-NP/GFP > Cherry-NP? Please make sure this is correct.

We have corrected this mistake and changed “GFP-NP/GFP > Cherry-NP” to mCherry-NP

-Line 288 (Fig.3c legend); c. > remove

We have done this

-Line 362; (white arrowheads and small dots) > (white arrowheads and small dots, Fig. 4e)

We have changed as suggested

-Fig. 5c; Sec31A > Sec31

We have unified the name Sec31A throughout the entire manuscript

-Line 424; GM310 > GM130

We have corrected this error

-Line 516 & 517; Fig. 8d > Fig. 8e, Fig.8e > Fig. 8d

To simplify, we have changed the elements order in Figure 8, so we wouldn't have to change the numeration throughout the text

-Line 535; c. > c. (bold)

We have done this

-Line 539 & 540 (Fig. 8 legend); swap d for e and e for d.

To simplify, we have changed the elements order in Figure 8, so we wouldn't have to change the numeration in the figure legend

Reviewer #2 (Remarks to the Author):

The paper significantly improved and the reviewer addressed all open issues.